# Antidiabetic Potential of Commonly Available Fruit Plants in Bangladesh: Updates on Prospective Phytochemicals and Their Reported MoAs

**DOI:** 10.3390/molecules27248709

**Published:** 2022-12-08

**Authors:** Safaet Alam, Anik Dhar, Muhib Hasan, Fahmida Tasnim Richi, Nazim Uddin Emon, Md. Abdul Aziz, Abdullah Al Mamun, Md. Nafees Rahman Chowdhury, Md. Jamal Hossain, Jin Kyu Kim, Bonglee Kim, Md. Sadman Hasib, S. M. Neamul Kabir Zihad, Mohammad Rashedul Haque, Isa Naina Mohamed, Mohammad A. Rashid

**Affiliations:** 1Drugs and Toxins Research Division, BCSIR Laboratories Rajshahi, Bangladesh Council of Scientific and Industrial Research, Rajshahi 6206, Bangladesh; 2Department of Pharmaceutical Chemistry, Faculty of Pharmacy, University of Dhaka, Dhaka 1000, Bangladesh; 3Department of Pharmacy, Faculty of Pharmacy, University of Dhaka, Dhaka 1000, Bangladesh; 4Department of Pharmacy, Faculty of Science and Engineering, International Islamic University Chittagong, Chittagong 4318, Bangladesh; 5Department of Pharmacy, State University of Bangladesh, 77 Satmasjid Road, Dhanmondi, Dhaka 1205, Bangladesh; 6Molecular Pharmacology Research Center, School of Pharmaceutical Sciences, Wenzhou Medical University, Wenzhou 325035, China; 7Department of Pathology, College of Korean Medicine, Kyung Hee University, Seoul 02447, Republic of Korea; 8Department of Pharmacy, Faculty of Allied Health Science, Daffodil International University, Dhaka 1216, Bangladesh; 9Department of Pharmacology, Faculty of Medicine, Universiti Kebangsaan Malaysia (The National University of Malaysia), Cheras 56000, Malaysia

**Keywords:** fruit, diabetes, insulin resistance, insulin sensitivity, α-glucosidase, α-amylase, HbA1c

## Abstract

Diabetes mellitus is a life-threatening disorder affecting people of all ages and adversely disrupts their daily functions. Despite the availability of numerous synthetic-antidiabetic medications and insulin, the demand for the development of novel antidiabetic medications is increasing due to the adverse effects and growth of resistance to commercial drugs in the long-term usage. Hence, antidiabetic phytochemicals isolated from fruit plants can be a very nifty option to develop life-saving novel antidiabetic therapeutics, employing several pathways and MoAs (mechanism of actions). This review focuses on the antidiabetic potential of commonly available Bangladeshi fruits and other plant parts, such as seeds, fruit peals, leaves, and roots, along with isolated phytochemicals from these phytosources based on lab findings and mechanism of actions. Several fruits, such as orange, lemon, amla, tamarind, and others, can produce remarkable antidiabetic actions and can be dietary alternatives to antidiabetic therapies. Besides, isolated phytochemicals from these plants, such as swertisin, quercetin, rutin, naringenin, and other prospective phytochemicals, also demonstrated their candidacy for further exploration to be established as antidiabetic leads. Thus, it can be considered that fruits are one of the most valuable gifts of plants packed with a wide spectrum of bioactive phytochemicals and are widely consumed as dietary items and medicinal therapies in different civilizations and cultures. This review will provide a better understanding of diabetes management by consuming fruits and other plant parts as well as deliver innovative hints for the researchers to develop novel drugs from these plant parts and/or their phytochemicals.

## 1. Introduction

Diabetes mellitus is a chronic metabolic disorder characterized by a lack of insulin action and/or generation. Discrepancies in protein, carbohydrate, and lipid metabolism can emerge due to the insufficiency of insulin [1]. Low insulin levels, insulin resistance in target tissues, insulin-receptor expression, especially in adipose tissue and skeletal muscles, and to a lesser extent in the liver, effector enzymes, and/or signal transduction system all can play vital roles in metabolic disorders [2]. The intensity of the symptoms has significant impact on the frequency and category of diarrhoea [1]. Diabetes is considered as one of the most frequent metabolic disorders worldwide, affecting roughly 2.8% of the global population, with a projected increase to 5.4% by 2025 [3]. Diabetes prevalence is also increasing in Bangladesh. In 2015, 7.1 million people had diabetes, while another 3.7 million cases went undiagnosed, and approximately 129,000 deaths were linked to the disease state, according to the International Centre for Diarrhoeal Disease Research in Bangladesh. Over the past two decades, the prevalence of diabetes has expanded 2.5 times, from 4.0% in 1995–2000 to 10.4% in 2010–2019 [1,3,4]. Though it is a non-transmittable illness, it has become one of the top five reasons for death in the world [4]. Some diabetic individuals, particularly those with type 2 diabetes, are asymptomatic throughout the early stages of the disease, while others exhibit apparent hyperglycemia. Due to ketoacidosis or rare non-ketotic hyperosmolar diseases, uncontrolled and unmonitored diabetes can result in stupor, coma, and even death if left untreated [5]. 

The interplay of hereditary and non-genetic variables may give rise to diabetes [1]. Despite the importance of diabetes categorization and its implications for treatment policy, it is quite ambiguous, and many diabetics, particularly younger adults, do not easily fit into one class [6]. Type 1, type 2, and gestational diabetes mellitus (GDM) are the standard classifications established by the American Diabetes Association (ADA) in 1997, and it is still the most widely accepted and recognized [5].

Many antidiabetic medicines are now commercially available to treat hyperglycemia, most of which function through improving insulin sensitivity, supplementing insulin, increasing insulin secretion, and stimulating glucose absorption. But these antihyperglycemic medicines, such as metformin and sulfonylureas, come with a slew of undesired side effects, including lactic acidosis and diarrhea (for metformin) and weight gain, liver failure, tachycardia, and hypothyroidism (for sulfonylureas) [7]. On the other hand, natural products are housed with a better safety profile along with cost effectiveness [8].

Herbal products, such as fruit, seed, bark, fruit peel, and leaf, are always considered as promising sources of bioactive phytochemicals to treat different ailments including diabetes, pain, fever, cancer, hypertension, and so on [9,10]. Phytomedicines are believed to be sanctified with lesser side effect, and thus, almost 80% of drug moieties are directly plant-extracted or their modified versions [11]. Fruits are one of the most notable natural sources which provide fiber, minerals, vitamins, and many other essential nutrients which are included in daily diets. Fruits are also rich sources of flavonoids, saponins, polyphenols, carotenoids, isothiocyanates, and several other bioactive phytochemicals [9,12]. Fruits are thought to be useful in the management of diabetes, cancer, obesity, and other disease states, including cardiovascular complications [9,13]. From ancient Chinese therapies to modern approaches, local fruits are heavily incorporated to treat diabetic patients [13]. Ayurveda medicines in the Indian subcontinent, including Bangladesh, also use a wide variety of locally produced fruits [13,14].

It is believed that fruits and other plant parts can exhibit antidiabetic potential through several mechanism of actions (Figure 1). Besides, other parts of these fruit producing trees, such as fruit peel, seed, leaf, and bark, can also alleviate several disease conditions including complications resulting from diabetes. Thus, this review work wasconducted in order to explore the antidiabetic potentials of commonly available Bangladeshi fruit plants (Figure 2) and provide unique insights towards the development of contemporary and novel therapies against diabetic states employing this locally popular fruit producing trees.

## 2. Traditional Uses of Fruit Plants in the Management of Diabetes

Fruits are the pivotal sources of vitamins, minerals, and several other phytochemicals consumed by people across the world as part of their daily diets. Several parts of fruit plants, including fruit, root, seed, leaf, and bark, are widely popular due to their medicinal properties in the management of several disease conditions. In developed countries, such as the US, Canada, Germany, Australia, and New Zealand, 20–25% of total drugs are made from medicinal plant parts, including fruits, while in fast developing countries, including India, Brazil, Indonesia, China and Russia, the ratio is skyrocketing to 80–85% [15]. The local use of fruit plants and other natural sources to treat diabetes is hugely promoted across the globe because of low cost, availability, and less side effects [16]. A vast number of plant parts and their fruits are used by the traditional healers of Kokrajhar district in India, Manisa, Turkey, and Urmia in Iran for their activity against diabetes [17,18,19]. Furthermore, a study reported that medicinal plants and their fruits were traditionally used for the treatment of diabetes in Manisa, Turkey [19]. *Aegle marmelos* (Bengal Quince), *Phyllanthus embelica* (Indian Gooseberry), *Carica Papaya* (Papaya) fruits are used in ethnic society of Bhopal region, Madhya Pradesh, India to treat diabetes [15]. Besides, in South Western Nigeria, many types of fruits are also used to treat diabetes mellitus, including lime (as lemonade) and orange (as lemonade as well as Infusion) [20]. Jujube dried fruits would be taken by ethnic people of Manisa, Turkey [19]. Juices from fruits and leaves are also popular in diabetes management i.e., *Citrus aurantifolia* (Key Lime) fruit juice along with decoction of *Magnifera indica* leaves are consumed by ethnic people of Agboville, Africa, while lime, orange, and coconut are consumed in the Dominican Community of New York City to treat diabetes mellitus [21,22]. In Nalbari district, Assam, India, leaves and fruits of Bengal quince as well as mango leaves are processed and consumed daily with cow’s milk [23]. Seed powder of muskmelon is used for its antidiabetic potentials in some areas as well [23].

## 3. Antidiabetic Potentials of Commonly Available Bangladeshi Fruits

### 3.1. Banana (Musa sapientum L.)

*Musa sapientum* L., a large-sized herbaceous monocot from the Musaceae family, is a well-known tropical Bangladeshi fruit and locally called ‘Kola’ [24]. The Kuk valley of New Guinea is the earliest occurrence of banana cultivation that has been documented. About 300 different types of bananas are cultivated in several regions, including Asia, Indo-Malaysia, and the Australian tropics [25]. It incorporates various bioactive phytochemicals, including anthocyanins, phenolic acids, flavanones, and terpenoids, which reportedly exhibit potential antidiabetic, anti-inflammatory, galactagogue, and antioxidant activities [26].

The lyophilized juice of *M. sapientum* stem (50 mg/kg dose) manifested antihyperglycemic action by enhancing insulin content in streptozotocin-induced diabetic rats [27]. The methanol extract of *M. sapientum* leaves at 250 and 500 mg/kg significantly reduced α-amylase activity by 79.6% in alloxan-induced male albino diabetic rats [28]. According to a report, the chloroform extracts of the flowers of *M. sapientum* displayed a notable antidiabetic effect (0.15, 0.20 and 0.25 g/kg b.w. dose) by lowering the glycosylated hemoglobin and blood glucose levels [25]. According to a research study, the aqueous extract of dried flowers of *M. sapientum* tends to exert antihyperglycemic activity in patients with type 2 diabetes at a dose of 5 mL/day [26].

### 3.2. Bengal Quince (Aegle marmelos (L.) Corrêa)

*Aegle marmelos* (L.) Corrêa, a slow-growing moderate-sized tree in the Rutaceae family, is a well-known subtropical Bangladeshi fruit and is locally known as ‘Bel’, ‘Bael’ [29]. It is widely found in several Asian countries, including India, China, Nepal, Sri Lanka, Myanmar, Pakistan, Bangladesh, and Vietnam [30]. Numerous bioactive phytoconstituents extracted from this plant are carotenoids, phenolic, alkaloids, pectins, tannins, coumarins, flavonoids, and terpenoids. These phytoconstituents reportedly exert an extensive spectrum of therapeutic actions, including antioxidant, cytotoxic, hypoglycemic, antimicrobial, hepatoprotective, anti-inflammatory, and cardioprotective potentials [31].

The methanol extract of bark of *A. marmelos* decreased the glucose content at 2 and 4 g/kg doses in streptozotocin-induced diabetic rats by 19.14% and 47.32%, respectively, which appears to be accomplished by rejuvenating pancreatic β-cells [32]. In the same rats at 120 mg/kg b.w. ip dose, the methanol extract of the leaves of *A. marmelos* also diminished the blood-sugar content by 54% [30]. Another in-vitro evaluation of the methanol extract of these leaves also showed notable antihyperglycemic action by hindering rat lens aldose reductase (RLAR) at an IC_50_ value of 15.00 ± 0.54 mg/mL [33]. According to a study, aqueous-leaf extracts of *A. marmelos* manifested hypoglycemic action(400 mg/kg b.w. dose) through lowering the blood glucose by 60.77% and improving the insulin content by 30.49% in alloxan-induced Albino Wistar diabetic rats [34].

### 3.3. Black Currant (Carissa carandas L.)

*Carissa carandas* L. is a deciduous thorny shrub of the Apocynaceae family [35]. Locally, *C. carandas* is known as “Koromcha” containing prospective bioactive phytochemicals [36]. *C. carandas* displayed notable antidiabetic, antioxidant, anti-inflammatory, antimicrobial, and antifungal properties [35].

Methanol extract of *C. carandas* leaves could sufficiently lower the increased blood- glucose concentration at 400 mg/kg dosage in alloxan induced diabetic rats [37]. Based on another study, methanolic extract of *C. carandas* fruit exhibited strong inhibition against α-amylase and β–glucosidase. The aqueous extract of *C. carandas* fruit has been demonstrated to have significant inhibitory activity against β–glucosidase, suggesting that it might be utilized as a sufficiently efficient treatment for postprandial hyperglycemia with minimum adverse effects [38].

### 3.4. Black Plum (Syzygium cumini (L.) Skeels)

*Syzygium cumini* (L.) Skeels from the Myrtaceae family, locally known as “Jaam”, “Kalajaam”, is a large-sized everlasting tropical tree [39] that is indigenous to the Indian subcontinent [40,41]. It is generally distributed throughout South Asia, including India, Indonesia, Sri Lanka, Bangladesh, Nepal, and a few other countries [41]. The chief bioactive phytochemicals of this plant are flavonoids, phenolic compounds, anthocyanins, carotenoids, essential oils, terpenes, and tannins, which reportedly demonstrate potential antidiabetic, anti-cancer, anti-inflammatory, antioxidant, and antimicrobial actions [42].

Black plum is a potent source of antidiabetic agents which can exert both antihyperglycemic and insulinotropic actions (Ayyanar et al. 2013, Teixeria et al. 2006). The ethanolic leaf extract of *S. cumini* is reported to exert in vitro antidiabetic action by halting α-glucosidase activity [43]. According to a report, the methanol and ethyl acetate extract of *S. cumini* seeds (400 and 200 mg/kg dose, respectively) tend to lessen the glucose content of blood in streptozotocin-induced diabetic rats [44]. The aqueous extract of the seeds of *S. cumini* manifests noteworthy hypoglycemic action in male albino-wistar rats (2.5 gm per kg body weight) by upraising the insulin secretion of pancreatic cells [45]. The methanol extract from the kernel fraction of *S. cumini* fruit displayed significant antihyperglycemic activity at an IC_50_ dose of 8.3 μg/mL by hindering α-amylase action by 98.2% [46]. Another research revealed that, high-fat diet-fed wistar rats with streptozotocin induced hyperglycemia were treated with the black-plum-aqueous-seed extract for 21 days which resulted in amelioration of insulin resistance and nourishment of the function β-cells [45,46].

### 3.5. Coconut (Cocos nucifera L.)

*Cocos nucifera* L. from the Arecaceae family, locally known as “Narkel”, is a long-lived single-stemmed tree and a renowned tropical Bangladeshi fruit. It has its origin in the Mexico, Brazil, Central America, India, Sri Lanka, and Indonesia [47]. Various bioactive phytoconstituents, such as phenols, tannins, leucoanthocyanidins, flavonoids, triterpenes, saponins, steroids, and alkaloids, have been extracted from this plant which exhibited a vast spectrum of pharmacological attributes, including bactericidal, anti-inflammatory, antineoplastic, antioxidant, hypoglycemic, anti-osteoporosis, and anthelmintic activities [48].

The hydro-methanol extract obtained from the spadix of *C. nucifera* tends to display remarkable antihyperglycemic activity in streptozotocin-induced diabetic Albino Wistar rats at 250 and 500 mg/kg body weight by escalating the insulin secreting power of the pancreatic β-cells [49]. According to a study, methanol extract of *C. nucifera* (200 mg/kg b.w. dose) manifested an antidiabetic effect by improving the content of not only sugar but also insulin in the blood stream in diabetic rats [48]. The extract obtained from *C. nucifera* husk using methanol tends to exert antihyperglycemic action in alloxan-induced diabetic rats having an IC_50_ range of 51.70 ± 4.66 μg/mL by significantly hindering the α-amylase activity [50]. The lyophilized coconut water of mature *C. nucifera* (1000 mg/kg b.w. dose) has been used to lower the glucose content from 275.32 ± 4.25 mg/dL to 129.23 ± 1.95 mg/dL in alloxan-induced male Sprague-Dawley rats [51].

### 3.6. Elephant Apple (Dillenia indica L.)

*Dillenia indica* L., locally known as “Chalta” in Bangladesh, is an evergreen tree [52] which belongs to the Dilleniaceae family [53] and mostly grows in the moist forests of sub-Himalayan region to Assam; it is a very familiar tree in household of Bangladeshi rural area [52]. *D. indica* has been reported to have phytochemicals, such as flavonoids, tannins and terpenoids, polyphenolic compounds, and saponins [54], which exert significant biological activities, namely antidiabetic, antimicrobial, antioxidant, dysentery, anti-inflammatory, and analgesic properties.

The extract of *D. indica* has shown the possible mode of antidiabetic action by enhancing the insulin impact by raising the insulin secreting potential of the pancreatic β-cells or its bound-state release or cell rejuvenation [52]. The methanol extract of *D. indica* demonstrated a notable antihyperglycemic action in alloxan and streptozotocin induced diabetic rats [53]. Furthermore, the methanol extracts of the leaves of *D. indica* enhanced the serum-insulin level by halting the function of α-amylase and α-glucosidase enzymes [55]. Based on a study, the leaves of *D. indica*, which have concentrations of 250 and 500 mg/kg b.w., were administered via the oral route and showed favorable impacts on the glucose content of blood [53].The alcohol extract of *D. indica* leaves given at doses of 100, 200, and 400 mg/kg for 45 days exerted remarkable depletion in the increased blood sugar content of rats at a fasting state (266.17 ± 7.07, 221.83 ± 5.70, 182.17 ± 3.59 mg/dL respectively). Consequently, the given doses cause a notable elevation in the insulin levels of serum (8.92 ± 0.15, 9.83 ± 0.13, 11.48 ± 0.39 mU/mL) [54]. The glucose content was prominently lessened in the blood by the methanol-leaf extracts of *D. indica* in streptozotocin-induced diabetic rats having the doses of 250 and 500 mg/kg [53].

### 3.7. Guava (Psidium guajava L.)

*Psidium guajava* L. is a large-sized everlasting tropical shrub belonging to the Myrtaceae family [56]. It is a popular pan-tropical fruit [57] and locally known as “Peyera” in Bangladesh. Though it is aboriginal to Central America, it is also found in southern Florida, Bermuda, and throughout the West Indies, from the Bahamas and Cuba through Trinidad and all the way south to Brazil [58]. It incorporates various bioactive phytoconstituents, such as saponins, alkaloids, tannins, cardiac glycosides, terpenes, flavonoids, and sterols. These compounds are likely to exert an extensive range of therapeutic attributes, including antidiabetic, antitumor, antimicrobial, antioxidant, and hepatoprotective activities [58].

The ethanol of stem bark of *P. guajava* tends to exhibit antihyperglycemic action (250 mg/kg oral dose) in alloxan-induced hyperglycemic rats [59]. In an in-vitro study, the ethanol and aqueous extracts of *P. guajava* leaf (1 mL concentration) halted the function of the enzyme alpha-amylase by 97.5% and 72.1%, respectively [60]. According to a study, ethanol extract of bark and leaf of *P. guajava* demonstrated hypoglycemic action by hindering α-glucosidase activity at an IC_50_ of 0.5 ± 0.01 and 1.0 ± 0.3 μg/mL, respectively [57]. The fruits also exhibited noteworthy glucose-diminishing activities in streptozotocin-induced diabetic rats. *P. guajava* reportedly safeguarded the pancreatic β-cells resulting from lipid peroxidation and DNA strand breakage mediated by streptozotocin and thus preserved insulin secretion. It also arrested the protein manifestation of pancreatic nuclear factor-kappa B caused by streptozotocin induction and attributed to its antihyperglycemic efficacy [61].

### 3.8. Hog Plum (Spondias mombin L.)

*Spondias mombin* L. is a rapid-growing perennial tree from the Anacardiaceae family. This is regarded as one of the renowned tropical fruits in Bangladesh, and is locally known as ‘Amra’. It is indigenous to America and Brazil, especially in the western Atlantic and Amazon forest [62]. The effective bioactive phytochemicals extracted from this plant are alkaloids, flavonoids, saponins, phenolic compounds, and tannins [63]. These phytochemicals reportedly exert a broad spectrum of notable pharmacological attributes, including anti-inflammatory, antioxidant, antidiabetic, antimicrobial, and antipsychotic activities [62].

The ethyl-acetate-soluble fraction obtained from the methanol extract of leaves of *S. mombin* manifested antihyperglycemic action in vitro by halting α-amylase and α-glucosidase with an IC_50_ level of 28.12 ± 0.48 μg/mL and 12.05 ± 0.002 μg/mL, respectively [64]. According to a study, methanol extract of leaves of *S. mombin* at concentrations of 200 and 400 mg/kg b.w. lowered blood sugar content by 20.03% and 33.33%, respectively in alloxan-induced diabetic male Wister rats. Consequently, in the same experimental animals, the applied doses displayed antidiabetic action by declining the glycosylated hemoglobin having concentrations of 7.72 ± 0.21% and 5.16 ± 0.09%, respectively [65].

### 3.9. Indian Goose Berry (Phyllanthus emblica L.)

*Phyllanthus emblica* L. (or *Emblica officinalis* Gaertn.), a prominent species from the Euphorbiaceae family is well-known as “Amlaki”, “Amla” in Bangladesh. This species is a medium-sized deciduous tree that has the height of about 8–18 m and is endemic to southeastern Asia, namely central India and Bangladesh [66]. *P. emblica* has been reported to have phenolic compounds, flavonoids, saponins, tannins, [67] alkaloids, proteins, glycosides, and amino acids [68], which exert a bunch of medicinal active properties, such as antidiabetic, hypolipidemic, antiatherogenic, antioxidative, anti-inflammatory, antibacterial, and cytotoxic properties [69].

Ethanol extract of *P. emblica* fruit (200 mg/kg b.w. for 45 days) exhibited substantial decrease in blood glucose and a notable rise in plasma insulin in streptozotocin-induced type 2 diabetic mice. Furthermore, *P. emblica* fruit extract inhibited α-glucosidase and α-amylase (IC_50_ values = 94.3 and 1.0 g m/L, respectively) [69]. *P. emblica* fruit extract also displayed positive hypoglycemic potency in an investigation conducted before [70]. Aqueous extract of *P. emblica* fruit, at a dosage of 200 mg/kg b.w., sufficiently reduced blood-glucose levels in alloxan-induced diabetic mice by suppressing gluconeogenesis and glycogenolysis [14]. The aqueous-methanol extract of its fruit resulted in a notable reduction in the blood-sugar level at fasting state, whereas uphill-serum insulin in diabetic rats, while also insulin-to-glucose ratio via rising β-cell size and number in diabetic rats, proving its antidiabetic activity through the upregulation of β-cell actions that decreases glucose intolerance and enhances insulin secretion [71]. According to a previous study, *P. emblica* fruit juice and hydroalcohol extract have rendered significant anti-diabetic action, as they could reduce blood glucose concentration and dramatically enhance glucose liberality in streptozotocin-induced type 1 diabetic rat models [72]. Another study demonstrated that oral administration of a hydroalcohol and methanol extract of *P. emblica* leaves culminated in a considerable suppression of fasting blood glucose and a rise in insulin content [73]. The aqueous extract of *P. emblica* fruit is proved efficient in lowering blood glucose and glycosylated hemoglobin (HbA1C), which were comparable to that of the antidiabetic-medication chlorpropamide [74]. Furthermore, fresh juice and hydroalcohol extracts of *E. officinalis* fruits were claimed to lower high-fastingblood-glucose levels, while increasing serum-insulin levels in streptozotocin-induced diabetic rats [74].

### 3.10. Indian Olive (Elaeocarpus floribundus Blume)

*Elaeocarpus floribundus* Blume, an evergreen medium-sized tree popularly known as “Jalpai” in Bangladesh, is a member from the Elaeocarpaceae family which is commonly found in Bangladesh, Madagascar, India, Southeast Asia, Malaysia, China, Japan, Australia, Fiji, and Hawaii [75]. *E. floribundus* has been reported to contain glycosides, flavonoids, steroids, terpenoids in fruits [76], phenolic acid, and anthocyanins, according to Zaman [77]. It displayed prospective pharmacological actions, such as antidiabetic, anticancer, antitumor, and antioxidant activities [75].

The methanol extract of *E. floribundus* leaf rendered noteworthy decrement of α-glucosidase enzyme having an IC_50_ value lesser in comparison to acarbose that establishes the antidiabetic activity of *E. floribundus* [75]. Based on another study, the stem bark of *E. floribundus* had also showed notable α-glucosidase inhibitory actions with an IC_50_ of 14.56 ± 1.20 µg/mL [78].

### 3.11. Indian Jujube (Zizyphus mauritiana Lam.)

*Zizyphus mauritiana* Lam. is a shrub belonging to the Rhamnaceae familyis, found from western Africa to India in the warm temperate zone [79], and widely cultivated in Bangladesh; it is also locally known as “Boroi”. *Z. mauritiana* has been shown to contain a variety of phytochemicals, including flavonoids, alkaloids, terpenoids, pectin, saponins, triterpenoic acids, lipids, and jujuboside saponins, which exerted potential sedative and hemolytic properties, sweetness-inhibitory effects and as an anxiolytic [79]. The cyclopeptide alkaloids were reported to have antibacterial, anticonvulsant, hypoglycemic, anti-infectious, diuretic, analgesic, antiplasmodial, and anti-inflammatory properties [80].

A study conducted on a hyperglycemic rat model with petroleum ether, chloroform, acetone, ethanol, aqueous, and crude aqueous extracts of *Z. mauritiana* fruits reveal to have antihyperglycemic action. The non-polysaccharide fraction of the aqueous extract of fruits of *Z. mauritiana* is said to have substantial antihyperglycemic and hypoglycemic effects [80]. Furthermore, aqueous and petroleum ether extracts of *Z. mauritiana* at 200 and 400 mg/kg dosages demonstrated significant antidiabetic effects [81]. The combination of aqueous and ethanol extracts of *Z. mauritiana* seeds (800 mg/kg extract of seeds and 10 mg/kg glyburide) improved glucose tolerance in both diabetic and normal mice. This finding implies the synergistic hypoglycemic action of *Z. mauritiana* extracts. According to the same study, aqueous and ethanolic extracts of *Z. mauritiana* seeds contain strong principles that may exert numerous activities involving several pathways to exert hypoglycemic and antihyperglycemic effects [82].

### 3.12. Indian Persimmon (Diospyros malabarica (Desr.) Kostel.)

*Diospyros malabarica* (Desr.) Kostel. is an intermediate evergreen shrub that may reach a height of 15 m, has dark grey or black bark, and exfoliates in rectangular scales [83,84]. Its fruits are locally known as “Gaab” which belongs to the Ebenaceae family. It thrives throughout the humid tropical climates of India and Bangladesh which constitutes phenols, tannins, proteins, flavonoids, alkaloids, and saponins [85]. *D. malabarica* has been reported to exert antioxidant, hypoglycemic, antidiarrheal, antiviral, antiprotozoal, anthelmintic, and cytotoxic activities [86].

Ethanol extract of *D. malabarica* bark restored the cell number and size of islet cells in diabetic rats. It also greatly enhanced the glucose tolerance test and blood glucose lowering action for up to 4 h [83]. Based on another research, methanol extract from *D. malabarica* fruits reduced fasting blood glucose, pancreatic thiobarbituric acid reactive compounds (TBARS), and serum lipid levels of alloxan-induced diabetic mice [84]. Furthermore, methanol extract of *D. malabarica* bark had a strong antihyperglycemic effect, resulting in an increased concentration of plasma protein and adrop in cholesterol and triglyceride levels [87].

### 3.13. Jackfruit (Artocarpus heterophyllus Lam.)

*Artocarpus heterophyllus* Lam. from the Moraceae family, is a medium-sized [88] everlasting monoecious tree locally known as “Kathal” [89]. Various studies revealed that A. heterophyllus was first introduced in the rain forests of the Western Ghats of Southwestern India though it also grows largely in Malaysia, Burma, Sri Lanka, Bangladesh, Indonesia, Philippines, and Brazil. [90]. It incorporates several bioactive phytochemicals like carotenoids, flavonoids, volatile acids, sterols, and tannins which exerted a vast array of therapeutic actions including, antioxidant, antidiabetic, antibacterial, and antitumorigenic properties [91].

The aqueous extract of *A. heterophyllus* fruit exerted antidiabetic action by impeding hemoglobin glycation with an IC_50_ value of 56.43% [92]. The hot water extract of leaves of *A. heterophyllus* displayed hypoglycemic action in the usual subjects and the diabetic patients at 20 g/kg equivalent dose by enhancing glucose tolerance [89]. According to a study, *A. heterophyllus* aqueous leaf extract enhances the glucose content of rat plasma in vitro, having concentrations from 125 to 2000 μg/mL by halting the function of α-amylase [90]. The ethyl acetate extract of leaves also showed notable antihyperglycemic action against streptozotocin-induced diabetic animal models (20 mg/kg b.w. dose) by enhancing insulin secretion of β-cells [91].

### 3.14. Java Apple (Syzygium samarangense (Blume) Merr. & L.M.Perry)

*Syzygium samarangense* (Blume) Merr. & L.M.Perry, a small-sized [88] evergreen tree which belongs to the Myrtaceae family [88], is a non-climacteric tropical Bangladeshi fruit [93] and locally known as “Jamrul”. Alongside Bangladesh, it is also available across Malaysia and also in the surrounding countries such as Thailand, Indonesia, and Taiwan [93]. It comprises several bioactive phytochemicals like phenols, flavonoids, flavonol glycosides, proanthocyanidins, anthocyanins, ellagitannins, chalcones, carotenoid, and triterpenoids which showed an extensive spectrum of potent therapeutic attributes, including antioxidant, anti-inflammatory, hypoglycemic, and antitumorigenic activities [94].

The methanol extracts of leaves of *S. samarangense* impeded serum glucose content by 59.3% in glucose-induced hyperglycemic mice at a 400 mg/kg b.w. dose [95]. Furthermore, according to a report, *S. samarangense* methanolic leaf extracts displayed antidiabetic action by impeding the alpha-glucosidase activity [96]. *S. samarangense* methanolic fruit extract showed hypoglycemic action in streptozotocin-induced diabetic rats at 100 mg/kg b.w. dose by effectively elevating the insulin-secreting power of the β-cells residing in the pancreas [94]. The aqueous extract of *S. samarangense* fruit tends to exert a remarkable antihyperglycemic action on the insulin-resistant FL83B mouse hepatocytes by effectively increasing glucose utilization and thereby improving the glycogen level [97].

### 3.15. Key Lime (Citrus aurantiifolia (Christm.) Swingle)

*Citrus aurantifolia* (Christm.) Swingle is a shrub that belongs to the Rutaceae family [98]. It has emerged from East Asian origins and more specifically, from northern Malaysia or India which is next to North Africa [99]. Locally, it is known as “Kagoji Lebu” and constitutes numerous phytochemicals among which pectins, flavonoids, and vitamins are biologically active. *C. aurantifolia* is likely to display promising antibacterial, antifungal, analgesic, anti-inflammatory, antioxidant, anthelmintic, [100] antidiabetic as well as antihyperglycemic potencies. The phytoconstituents present in *C. auirantifolia* are mainly flavonoids and coumarins which exerted antidiabetic actions [98]. In hyperglycemic rats, intraperitoneal administration of *C. aurantifolia* oil (100 mg/kg for 14 days) exerted a substantial decrease in fasting hepatic and blood glucose though the hepatic glycogen concentration was considerably enhanced [101].

Methanol and ethanol extracts of *C. aurantifolia* dried fruit have been reported to show substantial α-amylase inhibition and also being potent hypoglycemic agents [102]. Methanol extract of *C. aurantifolia* elevated TGF-β expression with an increased number of β-cells whereas LDL concentration and islets of Langerhans got dropped in hyperglycemic rats [103]. A methanol extract of its fruit peel demonstrated a progressive drop in fasting blood glucose volume and reduction in serum triglycerides in diabetic rats, establishing the antidiabetic potential of *C. aurantifolia* [104].

### 3.16. Lemon (Citrus limon (L.) Osbeck)

*Citrus limon* (L.) Osbeck from the Rutaceae family is a tiny, thorny, and evergreen tree attaining height of 10–20 feet and a native in Asian regions. In Bangladesh, *C. limon* is known as ‘Lebu’. The chief bioactive phytoconstituents that are secluded from both *C. limon* fruit and its juice are flavonoids, volatile oils [105], phenolic acids, coumarins, and amino acids [106]. Recently, notable therapeutic attributes of *C. limon* have been reported that include anti-inflammatory, antimicrobial, cytotoxic, and antiparasitic actions [106].

The extract of *C. limon* peel acquired using hexane remarkably lessened the glucose content in the bloodstream in alloxan-induced diabetic rats [107]. At 400 mg/kg daily dose via oral route for twelve days, the blood sugar range had been depleted notably by the ethanol extract of *C. limon* peels in streptozotocin-induced diabetic rats [106]. It can also inhibit the incidence of gluconeogenesis to prevent diabetic disorders [106].

### 3.17. Lotkon (Baccaurea motleyana (Müll.Arg.))

*Baccaurea motleyana* (Müll.Arg.) is an everlasting, deciduous tree that attains an altitude of 30 m. It is widely known as Rambai and as “Lotkon” in Bangladesh which belongs to the Phyllanthaceae family [108]. It is also found in Peninsular Malaysia to Sumatera, Borneo, and Halmahera [108]. *B. motleyana* contains phenols, flavonoids, fats, organic acids, and vitamins. [108,109]. Different parts of the Rambai tree were claimed to exert antibacterial, antihyperglycemic, and skincare properties [110]. Based on a previous study, the fruit of *B. motleyana* is quite beneficial for keeping blood sugar under control [108].

### 3.18. Lychee (Litchi chinensis Sonn.)

*Litchi chinensis* Sonn., a medium-sized everlasting subtropical tree from the Sapindaceae family [111], is said to be a non-climacteric type of Bangladeshi fruit [112]. It is locally famous as ‘Litchi’ [113]. In spite of its wide availability in Bangladesh, it is native to China’s Kwangtung and Fukien provinces and northern Vietnam [113]. This plant contains various effective bioactive phytochemicals such as sterols, coumarins, phenolic acids, chromanes, anthocyanins, flavonoids, lignans, sesquiterpenes, fatty acids, proanthocyanidins, and triterpenes [112]. *L. chinensis* is claimed to display potential anti-inflammatory, antioxidant, antidiabetic, and cytotoxic properties [114].

The crude extract of the seed of *L. chinensis* tends to exhibit antihyperglycemic action by hindering α-glucosidase activity at an IC_50_ of 0.691 μg/mL [115]. Again, in another study, *L. chinensis* seed water extract rendered hypoglycemic action in alloxan-induced diabetic rats by obstructing the glucose utilization of blood capillaries [116]. The pulp extract of *L. chinensis* demonstrated antidiabetic action with an IC_50_ of 10.4 mg/mL by halting alpha-glucosidase activity [112]. Based on a research study, an in vitro analysis of methanol and ethyl acetate extracts of *L. chinensis* exhibited noteworthy inhibitory actions on rat lens aldose reductase (RLAR) having IC_50_ values of 3.6 and 0.3 mg/mL, respectively [113].

### 3.19. Mango (Mangifera indica L.)

*Mangifera indica* L., a rapid-growing everlasting tree and also a member of the Anacardiaceae family, is said to be a renowned tropical Bangladeshi fruit [117] that is locally known as ‘Aam’ [118]. Despite having origins in semitropical countries like India, Bangladesh, and Myanmar, this is, however, still extensively farmed in the Philippines, Malaysia, Indonesia, Singapore, and Thailand [119]. Bioactive phytochemicals of this plant include saponins, flavonoids, terpenoids, steroids, tannins, anthraquinones, cardiac glycosides, and alkaloids. Based on previous reports, *M. indica* can render potential antidiabetic, antibacterial, antioxidant, and anti-inflammatory actions [119].

The ethanolic leaf extract of *M. indica* tends to exert notable antidiabetic action against streptozotocin-induced diabetic animal models (250 mg/kg b.w. dose) by elevating insulin secretion from β-cells [117]. According to a report, the extract of *M. indica* leaves obtained with ethanol normalizes blood glucose content at an IC_50_ dose of 2.28 mg/mL by impeding the activity of pancreatic α-amylase [119]. The alcohol extract of *M. indica* leaf can also display antihyperglycemic action in rabbits at dose concentrations of 50, 100, 150, and 200 mg/kg body weight [118]. The ethyl acetate fraction from *M. indica* methanol leaf extract manifested antihyperglycemic action in vitro by halting α-amylase and α-glucosidase enzyme at an IC_50_ of 24.04 ± 0.12 µg/mL and 25.11 ± 0.01 μg/mL, respectively [64]. According to a study, *M. indica* leaf extract rendered antidiabetic action in rats at an IC_50_ value of 1.45 mg/mL by hindering α-glucosidase activity [120]. An in vitro essay of *M. indica* methanol leaf extract exhibited notable hypoglycemic action by blocking the function of dipeptidyl peptidase-4 (DPP-4) with an IC_50_ value of 182.7 μg/mL [119].

### 3.20. Muskmelon (Cucumis melo L.)

*Cucumis melo* L. is an ancient herb that belongs to the Cucurbitaceae family [121]. This horticulture crop is found across the globe and widely cultivated in Bangladesh which is locally known as “Bangi”. *C. melo* contains enormous phytochemicals for example phenolic compounds, glycolipids, carbohydrates, flavonoids, and terpenoids following apocaretonoids which possess several biological activities including antidiabetic, antibacterial, anti-inflammatory, anti-hypothyroidism, antioxidant, and antiangiogenic activities [121].

The toluene soluble fraction of the ethanol extract of *C. melo* fruit was found to be highly effective in lowering blood glucose levels which are attributed to enhanced insulin secretion, suppression of glucose absorption from the gut, enhanced glucose absorption by adipose tissues and skeletal muscle, and ultimately reduction of glucose synthesis from hepatic cells [121]. Administration of *C. melo* leaf extract to streptozotocin-induced diabetic rats notably lessened the blood sugar together with glycated hemoglobin content [122]. On the basis of a previous report, seeds roasted at various temperatures of 150 °C, 200 °C, 250 °C, and 300 °C suppressed α-amylase at 2.0 mg/mL by 61.8 percent, 60.9 percent, 50.5 percent, 72.0 percent, and 45.7 percent, respectively. Furthermore, the hexane extract of *C. melo* seeds inhibited α-glucosidase activity significantly [123].

### 3.21. Orange (Citrus reticulata Blanco)

*Citrus reticulata* Blanco, a small-sized thorny everlasting tree from the Rutaceae family, is a widely popular fruit in Bangladesh and locally called ‘Komola’, ‘Kamala lebu’. Alongside Bangladesh, this fruit is grown worldwide and is also a horticultural crop in India [124]. It comprises various bioactive phytoconstituents like flavonoids, phenolics, tannins, monoterpenes, and sesquiterpenes and reportedly demonstrates a broad spectrum of remarkable antioxidant, antidiabetic, antifungal, anti-neurodegenerative, and antibacterial activities [124].

Orange peel and/or a combination of other citrus peels can exert potential action against diabetic disorders (Gosslau et al. 2018). The hydroethanolic extracts of *C. reticulata* fruit peel (100 mg/kg b.w./day dose) tend to exhibit hypoglycemic action in nicotinamide (NA)/streptozotocin(STZ)-induced type 2 diabetic rats by rejuvenating the function of β-cells found in the pancreas [125]. On the other hand, ethanol and aqueous peel extract manifested antihyperglycemic action by impeding the α-glucosidase by 70.8% and 14.8%, respectively, and also by hindering α-amylase 90.67% and 15.33%, respectively in rats with type 2 diabetes [126]. According to a report, the essential oil of *C. reticulata* rind and leaves (200 μL/kg b.w. dose) significantly lowered the blood glucose content from 251 ± 0.85 mg/dL to 90 ± 0.70 mg/dL, and 200 ± 0.67 mg/dL to 96.2 ± 0.86 mg/dL respectively in alloxan-induced diabetic rabbits [127].

### 3.22. Papaya (Carica papaya L.)

*Carica papaya* L., from the Caricaceae family, an ephemeral perennial [128] herbaceous laticiferous tree, is a popular tropical fruit and is locally known as “Pepe” [129]. It is widely available in Bangladesh, tropical America, southern Mexico, and neighboring Central America [128]. Several bioactive phytoconstituents like tannins, flavonoids, saponins, alkaloids, anthraquinones, cardiac glycosides, steroids, cardenolides, and phenolic compounds [130] have been reported from *C. papaya* which rendered a vast range of therapeutic attributes like antibacterial, anti-inflammatory, antiviral, hypoglycemic, and antitumorigenic activities [131].

The ethyl acetate extract of papaya seeds manifested antidiabetic action in vitro by impeding α-glucosidase and α-amylase enzymes at an IC_50_ of 83.54 and 36.84 mg/mL, respectively [132]. According to a study, the aqueous leaf extract (400 mg/kg b.w. dose) improved the blood sugar concentration in alloxan-induced diabetic albino rats [133]. In the streptozotocin-induced diabetic rats, aqueous extract of the leaves of *C. papaya* displayed antihyperglycemic action at a dose of 0.75 and 1.5 g/100 mL by upgrading the islet cells’ regenerative ability [134]. Consequently, the chloroform extract of *C. papaya* leaves also demonstrated remarkable hypoglycemic action in streptozotocin-induced diabetic rats [135].

### 3.23. Pineapple (Ananas comosus (L.) Merr.)

*Ananas comosus* (L.) Merr. is a monocot perennial plant and a member of the Bromeliaceae family. In Bangladesh, it is known as “Anarosh” in Bangladesh [136]. *A. comosus* consists of various effective bioactive phytoconstituents like alkaloids, acids, coumarins, flavonoids, glycoside, phenols, polyphenols, saponin, steroids, sterols, tacorin, tannins, terpenoids, and triterpenes [136]. It is claimed to exhibit an extensive spectrum of therapeutic actions like anti-hyperglycemic, anti-proliferative, anti-rheumatic, anti-inflammatory, antioxidant, antimicrobial, anti-coagulant, anthelminthic, anti-plasmodial, anti-pyretic, and cardioprotective properties [136].

The methanolic leaf extract of *A. comosus* leaves manifested a dose-dependent lowering of glucose content in the blood when given through an oral route to glucose-loaded Swiss albino mice models [137]. On the basis of a study conducted previously, the ethanol extract of *A. comosus* leaf notably halted the increase of glucose levels in the bloodstream having a concentration of 0.40 g/kg in diabetic rats. Also, the ethanolic leaves extract of *A. comosus* improved the sensitivity of insulin levels in rats with type 2 diabetes, which relates to enhancing the action of insulin in the hepatic cells [138]. Moreover, its fruit juice also showed synergistic action with glimepiride in lessening the blood sugar concentration in alloxan-induced diabetic rats [137].

### 3.24. Pomelo (Citrus maxima (Burm.) Merr.)

*Citrus maxima* (Burm.) Merr., an everlasting fragrant shrub from the Rutaceae family [139], is a famous pear-shaped Bangladeshi fruit [140] and is locally known as ‘Batabi lebu’, ‘Jambura’ [141]. This fruit is apparently endemic to South East Asia, India, and also many other tropical nations [140]. It incorporates various bioactive phytochemicals like phenolic acids, flavonoids, phytosterols, triterpenoids, saponins, and steroids which exerted potential biological activities like antidiabetic, antitumorigenic, hepatoprotective, anti-bacterial, and anti-hypercholesterolemic properties [139].

The peel extract of *C. maxima* manifested indispensable antihyperglycemic action in Alloxan-induced diabetic Wister rats (400 mg/kg body weight dose) by lowering the sugar content in the bloodstream by 70.17% [142]. Consequently, in the same model, the juice extract of *C. maxima* fruit improved the glucose content at 10 mL/kg b.w. dose [143]. With the help of an in vitro model, fruit juice of red *C. maxima* halted the functionality of both α-amylase and α-glucosidase enzymes by 79.75% and 72.83%, respectively, and thereby demonstrating antidiabetic action [140]. According to a report, the leaf extract obtained with methanol tends to exhibit a hypoglycemic effect in streptozotocin-induced diabetic rats when an amount of 200 and 400 mg/kg body weight is administered via an oral route [141].

### 3.25. Pomegranate (Puncia granatum L.)

*Punica granatum* L., often known as pomegranate, is a shrub that grows well in warm valleys and belongs to the Punicaceae family [144]. The fruit of the plant is locally known as “Dalim”. *P. granatum* is reported to have compounds such as alkaloids, tannins, flavonoids, anthocyanidins and hydroxybenzoic acid compounds [144]. It is efficacious as antidiabetic, antibacterial, anthelmintic, antifertility, antioxidant, antifungal, and antiulcer agents [145].

The aqueous extract of *P. granatum* fruit significantly raised the mRNA levels of IRS-1, Akt (Protein kinase B), GLUT-2, and GLUT-4, resulting in improved glucose uptake and storage and contributing to the regulation of both hyperglycemia and hyperlipidemia in alloxan-diabetic Wistar rats [146]. Ethyl acetate extract of its fruit peel halted α-glucosidase having an IC_50_ 285.21 ± 1.9 g/mL [147]. Furthermore, the methanolic flower extract of *P. granatum* increased cardiac PPAR-g mRNA expression and restored the down-regulated cardiac glucose transporter GLUT-4 (the insulin-dependent isoform of GLUTs) mRNA in rat models indicating that *P. granatum* flower extract has anti-diabetic activity due to improved insulin receptor sensitivity [148]. The methanolic extract of pomegranate fruit rinds also showed strong antidiabetic action in aldose reductase, α-amylase, and PTP1B suppression tests in a dose-dependent manner by controlling blood glucose concentrations within normal ranges in an alloxan-induced diabetes model [149].

### 3.26. Sapodilla (Manilkara zapota (L.) P.Royen)

*Manilkara zapota* (L.) P.Royen, well known as Sapodilla, is a medium to large tree which has its roots in the Indian subcontinent and is a species from the Sapotaceae family [150] that is locally known as “Sofeda”. Phytochemicals extracted from *M. zapota* are steroids, flavonoids, phenols, alkaloids, tannins, glycosides, and saponins which displayed enormous pharmacological effects like anti-diabetic, anti-arthritis, anti-inflammatory, anti-oxidant, anti-bacterial, anti-fungal, and anti-tumor activities [150].

Different phytochemicals from *M. zapota* seeds, leaves, and root extracts have been reported to exhibit hypoglycemic activity [150]. In addition, *M. zapota* leaf extract has shown improved hypoglycemic action in animal models [151]. Ethanol and aqueous extracts of *M. zapota* seeds displayed significant in vivo hypoglycemic action in experimental mice [152]. Another study found that alcohol and aqueous extracts of its seeds remarkably reduced the model group’s blood glucose levels when compared to metformin [153]. Furthermore, ethanol extract from the seed and methanol extract from the leaf of *M. zapota* decreased blood glucose levels by controlling insulin production from the few remaining β-cells. According to the findings of this study, enhanced peripheral glucose consumption also aided in the lowering of blood glucose levels [154].

### 3.27. Star Fruit (Averrhoa carambola L.)

*Averrhoa carambola* L., a slow-growing multi-stemmed tree from the Oxalidaceae family, is a popular nutrient-enriched fruit in Bangladesh and is locally called ‘Kamranga’ [155]. This plant is available in tropical areas such as India, Malaysia, Indonesia, and the Philippines [155]. Numerous bioactive phytochemicals extracted from this plant are saponins, flavonoids, alkaloids, tannins, phenols, anthocyanin and anthocyanidin, chalcones, aurones, catechins, and triterpenoids which exert a wide range of potential antioxidant, antidiabetic, antihypertensive, anti-hypercholesterolemic, anti-inflammatory, anti-infective, and cytotoxic activities [156].

The methanolic leaf extract of *A. carambola* (400 mg/kg b.w. dose) exhibited antidiabetic action with a blood glucose-reducing potential of 34.1% in glucose-filled Swiss albino mice [95]. According to a study, the hydroalcoholic leaf extract of *A. carambola* exhibited promising antidiabetic action by upgrading the glucose intake of muscles in male Wistar rats [156]. The ethanol extract of its bark manifested noteworthy antihyperglycemic action in vitro by effectively hindering the α-glucosidase enzyme at an IC_50_ value of 7.15 ± 0.06 μg/mL [157]. Besides, the ethanol extract from air-dried roots also demonstrated antidiabetic action by restoring the pancreatic β-cells function in streptozotocin-induced diabetic rats [158]. The juice of *A. carambola* fruits remarkably lowered glucose content at 25, 50, and 100 g/kg b.w. dose for three weeks in streptozotocin-induced diabetic mice [159].

### 3.28. Sugar apple (Annona squamosa L.)

*Annona squamosa* L., a small, evergreen, semi-deciduous tree from the Annonaceae family [160], is well-known nutritious tropical Bangladeshi fruit and is locally known as ‘Ata’ [161]. It is also found in India, the West Indies, southern America, and Thailand [161]. It comprises numerous bioactive phytochemicals like volatile Oils, alkaloids, terpenoids, flavonoids, polyphenols, glycosides, saponins, and tannins. Based on previous reports, A. squamosa can exhibit promising antioxidant, cytotoxic, anti-inflammatory, hypoglycemic, antimicrobial, hepatoprotective, and antiplasmodial actions [161].

*A. squamosa* hydroalcoholic leaf extract (350 mg/kg b.w. oral dose) tends to exert antidiabetic action with glucose reducing activity by 50.11% in streptozotocin-induced diabetic rats [162]. An in vitro analysis of methanol extracts of barks and leaves of *A. squamosa* manifested hypoglycemic action by impeding α-amylase at an IC_50_ value of 123.91 and 153.89 μg/mL, respectively [163]. The hexane extract of the leaf of *A. squamosal* obtained using reportedly exerted antihyperglycemic action in vivo having a concentration of 500 mg/kg p.o by hindering the human PTP1B enzyme with an IC_50_ level of 17.4 μg/mL [164]. Its ethanolic leaf extract at 50 mg/kg dose lessened the blood sugar content in alloxan-induced diabetic rabbits by 52.7% [161]. The ethanol and methanol extracts of *A. squamosa* seeds (200 mg/kg b.w. dose) demonstrated a notable antihyperglycemic effect by reducing the blood sugar levels by 43.96% and 45.99%, respectively, in alloxan-induced diabetic rats [165].

### 3.29. Watermelon (Citrullus lanatus (Thunb.) Matsum. & Nakai)

Watermelon, scientifically known as *Citrullus lanatus* (Thunb.) Matsum. & Nakai is the most commonly grown species from the Cucurbitaceae family. It is widely farmed throughout the globe, including Bangladesh [166], and is locally known as “Tormuj”. This plant’s bioactive phytochemicals are mostly ascribed to triterpenes, sterols, and alkaloids which can be reportedly used as antidiabetic, anthelmintic, diuretic, antibacterial, antifungal, and antihypertensive agents [167].

Previous research has revealed that the ethanol extract of *C. lanatus* leaf can inhibit pancreatic α-amylase with an IC_50_ value of 36.75 ± 3.47 g/mL [168]. The aqueous ethanol and aqueous extracts of *C. lanatus* leaves profoundly rendered anti-diabetic action via α-glucosidase suppression, with IC_50_ values ranging from 26.26 ± 0.29 to 180.33 ± 1.31 g/mL [166]. In another study, its juice was also discovered to have promising anti-diabetic efficacy in an experimental diabetic animal model via several mechanisms involving the regulation of glucose transporters and domination of α-glucosidase and α-amylase activity [168]. The seed extracts of yellow-skinned *C. lanatus* hindered α-glucosidase with IC_50_ values ranging from 32.50 ± 0.36 to 313 ± 1.36 g/mL for 70 percent aqueous ethanol and aqueous extracts, respectively [166]. A recent study also found that the alcalde and tryptic hydrolysates of *C. lanatus* seeds had a very strong α-amylase inhibitory capacity (IC_50_ values of 0.149 to 0.234 mg/mL) via a non-competitive suppression mechanism [169].

### 3.30. Tal Palm (Borassus flabellifer L.)

*Borassus flabellifer* L., a famous species of the Arecaceae family is locally known as “Tal” in Bengali and is a tall palm reaching up to 12–33 m height having a black stem and crown of leaves at the top [170]. It is extensively disseminated and grown in tropical Asian nations like Thailand, India, Myanmar, Sri Lanka, Malaysia, and Bangladesh. It is a promising source of alkaloids, flavonoids, glycosides, tannins and so many phenolic compounds [170]. *B. flabellifer* reportedly exerts an extensive spectrum of therapeutic actions like antihyperglycemic, anti-inflammatory, antipyretic, antibacterial, and anthelmintic activities [170].

Ethanolic extract of tal palm roots at 250 and 500 mg/kg displayed a promising reduction in the glucose content of serum in rats with type 2 diabetes [171]. Numerous portions of *B. flabellifer* like leaves, roots, pulp, and fruit fibers. are claimed to be utilized to treat diabetes [170]. Ethanolic extract of the flower of *B. flabellifer* significantly improved the glucose tolerance up to 4 h and lowered the glucose content of blood significantly in alloxan-induced diabetic rats. Furthermore, ethanol extracts of *B. flabellifer* flowers showed notable regeneration of pancreatic β cells comparable to glibenclamide [172]. Methanol extract of tal palm fruits demonstrated raised content of insulin in the plasma when compared to diabetic controls in diabetic rats. Again, methanol extract from tal palm fruits also impeded PTP1B remarkably with an IC_50_ value of 23.98 mg/mL, and this decreased PTP1B expression which might increase the mass of β-cells in the pancreas which subsequently improved the release of insulin triggered by glucose and ultimately lowered the glucose content in blood [173].

### 3.31. Tamarind (Tamarindus indica L.)

*Tamarindus indica* L. from the Fabaceae family, a large-sized everlasting tree, is a nutrient-enriched tropical fruit is locally named “Tetul”. Though this fruit is mostly endemic to tropical regions of Africa, still tamarind is farmed and developed well in all other tropical continents. *T. indica* is reported to have phytoconstituents like flavonoids, tannins, glycosides, organic acids, and phenolic compounds [174]. *T. indica* is reported to be useful as an antidiabetic, antimicrobial, and anti-inflammatory agent [175].

The aqueous extract of *T. indica* fruit pulp showed promising postprandial hypoglycemic effect by impeding the function of α-amylase and α-glucosidase enzymes and raising the glucose uptake [176]. Aqueous extract from *T. indica* seed and seed coat showed improvement in hyperlipidemia, which is foundnotably in rat models and humans [175]. On the basis of a study conducted previously, the aqueous extracts of *T. indica* seeds exerted hypoglycemic activity which was mediated by lowering the diffusion rate of blood glucose, increasing glucose adsorption, and upgrading the transportation of blood glucose at the cellular level throughout the plasma membrane [177]. According to a report, the extract of *T. indica* seed obtained using methanol demonstrated a significant reduction of blood glucose content at the fasting state in mice models [178]. In addition, the alcoholic extract of its stem bark functioned as a potent hyperglycemic agent in the treatment of diabetes mellitus [179]. The ethanolic fruit pulp extract also effectively changed alloxan-induced alterations in serum glucose, enzyme, and lipid profile [180].

### 3.32. Wood Apple (Limonia acidissima Groff)

*Limonia acidissima* Groff, a slow-growing, fragrant, large-sized, deciduous tree from the Rutaceae family is a popular nutrient-enriched Bangladeshi fruit and is locally known as “Kod-bael” [181,182]. Bangladesh, India, Pakistan, Sri Lanka, Myanmar, and Vietnam are among the countries where it appears to be growing naturally [182]. It incorporates major bioactive phytochemicals like coumarins, lignans, flavonoids, phenolic acids, quinones, alkaloids, triterpenoids, sterols, and volatile oils which reportedly exert a vast range of therapeutic attributes like antioxidant, cytotoxic, hypoglycemic, antimicrobial, and hepatoprotective activities [183].

The aqueous and ethanol extracts of *L. acidissima* stem bark (200 mg/kg b.w. dose) remarkably lessened the blood glucose content from 250–358 mg/kg to 99.8 and 112.6 mg/kg, respectively in alloxan-induced diabetic rats [181]. According to a report, the methanolic fruit extract of *L. acidissima* tends to exhibit antidiabetic action by lowering the glucose content by 39% and 54.5% at an oral dose of 200 and 400 mg/kg b.w., respectively in streptozotocin-induced male Albino diabetic rats [184]. The methanol and aqueous extract of its fruit manifested antihyperglycemic action in vitro (100 μg/mL concentration) by impeding the α-glucosidase at an IC_50_ of 66.738 and 84.548 μg/mL, respectively and also by hindering α-amylase at an IC_50_ of 119.698 and 167.505 μg/mL, respectively [185].

## 4. Phytochemicals from Fruits and Other Plant Parts

Phytochemicals are the key factors that exert the pharmacological actions of medicinal plants [9,11]. Thus, the presence of notable phytochemicals can explain the pharmacological properties of plants such as antidiabetic, anticancer, antidiarrheal, and antihypertensive activities. Several types of phytochemicals including tannins, saponins, flavonoids, glycosides, and phenolic compounds are very promising agents against diabetic complications. Phytochemicals with antidiabetic potentials isolated from these aforementioned species have been presented in Table 1 along with their mechanisms of action. Besides, prospective phytochemicals corroborated with antidiabetic potentials have been also classified according to chemical classes in Table 2. 

## 5. Clinical Trials of Prospective Fruit Plants and Their Phytochemicals to Treat Diabetes Mellitus

Despite the fact that fruit plants are excellent sources for the treatment of a variety of diseases, including diabetes, many plants have not been thoroughly researched clinically yet. Some plants have been subjected to clinical trials which actually provide hints about the immense potential of fruit plants in the management of diabetic complications. Based on previously reported studies, the administration of 4–24 g of the powder of black plum seed to 28 diabetic individuals reportedly showed hypoglycemic action through a diminution in the mean fasting and postprandial glucose contents in the blood. Besides, a considerable antihyperglycemic effect was seen in thirty individuals with uncomplicated type 2 diabetes due to the ingestion of 12 g powder of black plum seed for a course of three months in three divided doses [262]. Again, the aqueous extract of guava leaf tends to exert antidiabetic action in twenty hospitalized non-insulin-dependent diabetes patients by declining the postprandial blood glucose content from 160 mg/dL to 143 mg/dL [263]. In patients with non-insulin-dependent diabetes, the extract tablets of lychee seed (30 g/day) for a duration of 12 weeks exhibited a hypoglycemic effect by lowering the fast plasma glucose content. In the same patients, the blood sugar level was reportedly lower at a concentration of 3.6–5.4 g/day after the administration of lychee seed extract tablets [264]. According to another study, the oral administration of 5 g of the leaf of Bengal quince for a duration of one month in 10 patients with type 2 diabetes exerted significant hypoglycemic action by declining the pre and postprandial blood sugar levels. Moreover, in 4 groups of a total of 120 type 2 diabetes subjects, the ingestion of 2 g of leaf powder and 2 g of the combination of pulp and seed powder of Bengal quince for 3 months tends to show a substantial lowering in fasting blood sugar content [29]. Furthermore, a double-blind, randomized, controlled clinical trial that lasted 8 weeks and involved 52 obese type 2 diabetic patients (26 men and 26 women; ages 30 to 50) discovered that administration of P. granatum supplements significantly decreased fasting blood glucose from 161.46 mg/dL to 143.50 mg/dL. Additionally, the patients’ GLUT-4 gene expression increased [265]. Another single-blind, randomized-controlled clinical study on 44 type 2 diabetic patients (age range, 56–6.7 years; 23 men, 21 women) discovered that P. granatum juice significantly reduced oxidative stress, indicating that consumption of it could delay the onset of oxidative stress-related diabetes mellitus [265]

Phytochemicals have also been subjected to clinical trials to discover new sources of antidiabetic medications. Based on previous clinical trials on 36 patients with non-insulin-dependent diabetes, 500 mg of berberine three times a day is likely to lower the fasting blood glucose level from 10.60 ± 0.9 to 6.9± 0.5 mmol/L in a three-month time frame. According to another study, 500 mg of berberine along with 5 gm of each glipizide and metformin in 60 type 2 diabetes patients is reported to enhance the state of glucose metabolism and insulin sensitivity by lowering the fasting blood glucose content [266]. In 45 patients with non-insulin-dependent diabetes, the administration of hesperidin at 500 mg/day for 8 weeks can lead to an increment in insulin content and a potential reduction of fast blood glucose content [267]. The administration of 180 mg ellagic acid capsules in 22 type-2 diabetes patients for 8 weeks remarkably reduced the fasting blood sugar content and thereby showed antihyperglycemic action [268]. In 33 patients with non-insulin-dependent diabetes, pinitol at 400 mg thrice a day for a tenure of 3 months is reported to exert antidiabetic action by declining the fasting plasma glucose content and enhancing the secretion of insulin [269]. Cetrulline, another prospective phytochemical, is reported to exhibit notable antihyperglycemic action by remarkably lessening the fasting plasma glucose content at an oral administration of 3 g per day for 8 weeks in 23 patients with type-2 diabetes [270].

## 6. Reported Mechanism of Actions to Exert Antidiabetic Potentials

**a.** 
**Inhibition of α-glucosidase secreted from the brush border of the small intestine**


Mammalian α-glucosidase is a membrane-bound hydrolytic enzyme, located in the epithelia of the small intestine’s mucosal brush border, which facilitates carbohydrate digestion. Inhibitors of this enzyme prevent carbs from being cleaved, resulting in less glucose absorption and a lower postprandial glycemic level [271,272].

**b.** 
**Inhibition of DPP-4 enzyme**


Incretin hormones include glucagon-like peptide-1 (GLP-1) and glucose-dependent insulinotropic polypeptide (GIP), which facilitate the secretion of insulin. A serine peptidase enzyme called dipeptidyl peptidase-4 (DPP-4) breaks down these hormones quickly. Hence, inhibitors of the DPP-4 enzyme have anti-diabetic properties by stimulating insulin secretion and inhibiting glucagon secretion [273,274].

**c.** 
**Inhibition of α-amylase secreted from the salivary gland**


Inhibition of the enzyme, α-amylase, which is found mainly in saliva and pancreatic juice, can lead to lower postprandial blood glucose levels. As it breaks down starch and glycogen and increases the blood sugar level. Hence, this enzyme’s inhibition helps to control diabetes [275].

**d.** 
**Increased secretion of insulin**


An increase in intracellular calcium ion [Ca^2+^]_i_ stimulates pancreatic β cells and facilitates insulin secretion. Some phytochemicals, e.g., p-methoxy cinnamic acid acting on the L-type Ca^2+^ channels have been demonstrated to boost insulin release by increasing cAMP via the inhibition of phosphodiesterase [276,277].

**e.** 
**Increased insulin sensitivity and improved glucose uptake by muscle cells and adipose tissue**


The sensitivity of non-pancreatic cells to insulin is enhanced by certain phytochemicals, resulting in better glycemic management. Glucose uptake is increased in skeletal muscle and adipose tissue due to the activation of a sequence of processes that occur in response to a rise in insulin levels. Insulin promotes the phosphorylation of protein substrates and increases the uptake of circulating glucose by adipose tissue and muscle cells when it interacts with insulin receptors [278].

**f.** 
**Nourishment of Pancreatic β-Cells**


Insulin-secreting pancreatic β cells can be impaired by autoimmune processes mediated via macrophages, cytokines, and T cells weaken them in type 1 diabetes and by oxidative stress, elevated lipid or glucose levels, and inflammatory mediators in type 2 diabetes. They can be strengthened against reactive oxygen species accumulation and lipid peroxidation-mediated cell death by increasing antioxidants, such as reduced glutathione (non-enzymatic) and catalase, superoxide dismutase, glutathione peroxidase, glutathione S transferase (enzymatic) [279,280].

**g.** 
**Reduction of HbA1c and glycated plasma protein levels**


In diabetes mellitus, blood glucose content is increased and monosaccharides react non-enzymatically with blood proteins (mostly hemoglobin A and albumin) in a process known as glycation. Glycation inhibitors obstruct this process by a variety of methods, including competitive interaction with the protein’s amino group, cleaving the open chain of monosaccharides, binding at the glycation site, and attaching with the intermediates of glycation reaction. As a result, HbA1c and glycated plasma protein concentrations are reduced, and the consequences of glycation and diabetes problems can be avoided [281,282,283].

**h.** 
**Improvement of Glucagon-like peptide-1 (GLP-1)**


GLP-1 (glucagon-like peptide-1) is a hormone produced by L cells in the gastrointestinal system’s distal ileum and colon. It slows stomach emptying, suppresses hunger, and imparts a sense of fullness by increasing glucose-dependent insulin secretion and decreasing glucagon release. Alternative drugs that operate as agonists for the gluco-protein-coupled receptor (GLp-1R) have been identified as viable options for achieving the desired effect. They boost insulin production via raising insulin gene transcription and intracellular Ca^2+^ levels, as well as activating the pancreatic duodenal homeobox 1 transcription factor that promotes insulin gene expression (Pdx-1) [284,285].

**i.** 
**Regulation of Glucose transporter type 4 (GLUT-4)**


Glucose transporter type 4 (GLUT-4) is a transporter with a 12-transmembrane domain that facilitates insulin-induced glucose influx into skeletal muscle and fat cells. The transporter is normally found intracellularly, but it moves to the cell membrane in response to insulin stimulation or during exercise via separate processes. Insulin receptor (IR) tyrosine kinase is activated when insulin binds to its receptor in target cells, triggering phosphorylation-mediated activation of other protein kinases that finally mobilize the effectors, especially Rab proteins [265,286].

## 7. Pharmacokinetic and Toxicological Profiles of Phytochemicals

Several studies have evaluated the pharmacokinetic data of phytochemicals isolated from plant sources i.e., natural alkaloids such as aegeline. The pharmacokinetic property of aegeline was demonstrated by Manda et al. [287]. Experimental mice were administered an oral dose of aegeline (30 mg/kg) that produced an elimination half-life of 1.25 h. Moreover, the area under the curve (AUC) and the peak plasma concentration were 2 h × µg/mL and 0.92 µg/mL, respectively. The volume of distribution (V_D_) was found to be 40 L, and the distribution was reported in the kidney, brain, and liver of the mice. The information is found to be quite significant when comparing the outcomes of the in vitro cytotoxicity test. Aegeline, extracted from the fruit and leaves of natural plants, was experimented on a human liver carcinoma cell line (HepG2) and it was found that the highest dose tested (i.e., 100 ug/mL or 336.3 µmol) caused 43.2% inhibition but failed to reach an LC50 or IC50 [288]. Multiple investigators also tested the pharmacokinetic properties of berberine which demonstrated that berberine undergoes extensive metabolism resulting in extremely low oral bioavailability of this compound [289,290,291,292,293]. The inconsistency between the pharmacological activity of berberine and its extremely low plasma bioavailability proves that the berberine metabolites may influence its bioactivities [294,295,296]. In in-vitro tests on human liver microsomes using coumarin showed that the elimination rate was quite rapid. This is due to the 7-hydroxylation of the compound by cytochrome (CYP) P450 2A6 [297] and 3,4-epoxidation by CYP P450 2E1 [298]. On the other hand, 3,4-epoxidation of coumarin in rat liver microsomes revealed that coumarin is slowly eliminated by forming o-hydroxyphenyl acetic acid as the main metabolite [299]. The pharmacokinetic properties of phenolic compounds and tannins have been tested in both human and animal studies [300] but the results are controversial. A study by Wiese et al. [301] reported that high molecular weight tannins do not absorb intact, but rather converted to other metabolites. However, high fructose and sucrose contents may lead to type 2 diabetes mellitus along with cardiovascular risk [302]. Thus, fruits with high fructose and sucrose level should be avoided to avoid further triggers of diabetic complications. Besides, overconsumption of citrus fruits can cause toxicity due to their essential oil content [303]. Starfruits can induce both nephrotoxicity and neurotoxicity attributed to oxalate and caramboxin content. This fruit thus should not be consumed during renal impairment and empty stomach. Moreover, the calcium carbide used in the fruit ripening process can cause skin burns, skin irritations and inflammation. Other impurities found in calcium carbide including arsenic along with other toxic and carcinogenic chemicals can worsen the situation. On the other hand, canned fruit juices can also exert toxicity based on tin levels of around 1400 ppm and above. Thus, consumption of fruits should not be excessive which can lead to problems like gastric discomfort, a spike in glucose level or other complications.

## 8. Discussion

Type 1 and type 2 diabetes mellitus are the two most common types of diabetes, both of which cause hyperglycemia. Type 1 diabetes mellitus is an autoimmune disease defined by the destruction of pancreatic cells as a result of a very severe insulin deficiency state. Type 2 diabetes mellitus, on the other hand, is more well-known, and it affects 90 to 95% of total diabetic individuals. It is defined by peripheral insulin resistance and insulin secretion abnormalities [304]. Despite the fact that diabetes is a non-communicable disease, experts warn that by 2030, there will be about 438 million diabetic individuals [7], demonstrating the severity of the disease. Insulin resistance, aberrant insulin secretion, and hepatic glucose production, as well as poor lipid metabolism, are all factors that contribute to diabetes. Insulin resistance is a condition in which the potency of insulin on target cells, particularly adipocytes, hepatocytes, and skeletal muscles, is breached [305], resulting in hyperglycemia by interfering with glucose utilization and enhancing hepatic glucose output [306].

Diabetes is usually treated and controlled by prescribing synthetic antidiabetic drugs such as biguanides (metformin), sulphonylureas (glibenclamide), thiazolidinediones (pioglitazone), alpha-glucosidase inhibitors (acarbose), DPP-4 inhibitors (sitagliptin), glinides (repaglinide) and GLP-1 agonists (exenatide) [7]. Despite their widespread use, synthetic drugs have numerous side effects that gradually limit their prescription and welcome extensive research on the discovery and development of plant-based antidiabetic therapies with improved safety and efficacy profiles [307,308]. Diabetes mellitus has also been treated with plant-based drugs due to their efficacy, minimal toxicity and side effects, low cost, and availability in several localities [309,310]. To bring ethnobotanically important plants to the commercial field, isolated phytochemicals can be subjected to extensive research and clinical trials in order to validate the scientific potential of tribal or local use [311,312]. For example, in the 25 years prior to 2007, about 50% of all approved medications were plant-based natural compounds or their synthetic analogs [308,313]. The widely used diabetes medicine metformin is also produced from a plant source, *Galegine officinalis* [314]. Phytochemicals with antidiabetic potential include flavonoids (quercetin), alkaloids (berberine), terpenes (thymoquinone), phenolic compounds (6- gingerol), tannins (punicalagin), and others. Based on previous studies, phytochemicals can exert potent antidiabetic effects by enhancing insulin sensitivity and secretion, increasing glucose absorption by muscle and adipose tissue, and nourishing pancreatic beta cells. However, more research is required to determine these compounds’ exact mechanisms of action, allowing them to be developed as medications or chemical leads. Phytochemicals not only act as medications or drug templates but also help uncover complex and novel biochemical pathways and targets involved in the disease [315]. Thus, additional research on these phytochemicals may reveal many treatment targets for diabetes mellitus, though the determination of the feasibility and toxicity profile of plant-based products is also a major research concern [7]. A wide range of fruit plants containing prospective phytochemicals with antidiabetic potential is commonly found in Bangladesh and the locals are consuming those fruits and other plant parts daily with or without knowing their health benefits (Figure 3). Fruits are packed with several bioactive secondary metabolites which may attribute to their antihyperglycemic activity [13]. Choosing fruits as antidiabetic agents can be very efficient as we generally consume fruits every day and if our daily consumption can support us to control and/or prevent diabetes, it can be at the same time cost-effective and widely available option. Besides, other fruit parts like seeds of black plum and litchi, peels and leaves of lemon, and flowers of the pomegranate can be equally prioritized in the management of diabetes. Moreover, reported antidiabetic phytochemicals from these sources (fruits and other plant parts) like swertisin, quercetin, berberine, hesperidin, kaempferol, 6-gingerol, ellagic acid, pinitol, and other prospective phytochemicals should be evaluated extensively to unveil novel therapeutics against diabetic disorders. Fruits and other parts of plants are also addressed in many indigenous medicinal applications, including in ancient Chinese cultures, Ayurveda, and Western medical therapies which also indicate their enormous potential to be a great source of notable antidiabetic drugs [7,13]. There have also been some previous studies on the antidiabetic potential of fruit plants. But in our review work, we exclusively stratified the phytochemicals according to their source narrowed down to plant parts along with their reported MoA (mechanism of action). Thus, we hope that, this review containing a balanced scoop of information of fruit plants and their antidiabetic importance can be crucial in the field of novel antidiabetic drug development. Even, consumption of fruits can be encouraged by this scientific review in order to manage diabetic complications among local people.

## 9. Materials and Methods

### Article Search Strategy

An extensive literature search was performed employing PubMed/Medline, Scopus, ScienceDirect, Web of Science, Google Scholar, and Wiley Online Library databases. We have utilized ‘Antidiabetic activity’, ‘Diabetes’, ‘Type 2 diabetes’, ‘T2DM’, ‘Fruits’, ‘Fruit plants’, ‘Bangladesh’, ‘Bangladeshi fruit’, ‘Fruit extracts’, and ‘Plant product’ keywords to collect desired articles up to April 2022. Taking into account only peer-reviewed and published publications in the English language on the antidiabetic actions of commonly available Bangladeshi fruit plants and their isolated phytocompounds as inclusion criteria, 323 distinct articles were included in this review work. The PRISMA (was followed to search, extract, and include necessary information for this article (Figure 4) [316].

The following were the inclusion criteria of studies: (a) The study must evaluate the antidiabetic activity of the fruit plants; (b) The plants must be available in Bangladesh; (c) The study has comprehensively discussed the potential antidiabetic activities as well as cited relevant sources; (d) The data including the values presented was authentic; and (e) The study was published in the above-mentioned authentic databases.

Besides, the exclusion criteria were as follows: (a) Plant species other than fruit trees are discarded; (b) Not commonly grown fruit plants of Bangladesh are discarded; (c) No reported antidiabetic potential from the fruit plants are discarded; (d) Fruit trees with no reported phytochemical with antidiabetic action are discarded; (e) Articles not in the English language are discarded; (f) Commentary/letter to editor/correspondence are discarded; (g) Fruits with only traditional use but lack of scientific reports have been discarded; and (h) Not peer-reviewed and/or published data are discarded.

## 10. Conclusion and Future Perspectives

Though fruits and other plant parts have long been thought to be viable sources of treatments for a variety of diseases, including diabetes and others, many of those have yet to be thoroughly investigated. This review sought to investigate available Bangladeshi antidiabetic fruit-producing trees that have been shown in lab studies to have antihyperglycemic properties and may be used to treat a variety of diabetic and metabolic illnesses. Based on the review work, fruit trees can deliver a handy source of diabetes management blessed with a better safety profile while a major number of synthetic medications are compromised with side effects including weight gain, liver failure, tachycardia, and hypothyroidism as well as a higher cost. Prospective phytochemicals like quercetin, rutin, naringin, 6-gingerol, pinitol, and others may play the driving role here to exert the reported antidiabetic potential. By inhibiting the α-glucosidase, α-amylase, and DPP-4 enzymes, along with amelioration of insulin sensitivity, insulin secretion, glucose uptake by muscle cells and adipose tissue, and nourishment of pancreatic β -cells, the phytochemicals reported in this study demonstrated prominent anti-diabetic action. These various modes of phytochemicals to exert antidiabetic action illustrates the effectual diversity they can offer. Therefore, prospective and potential candidates for the development of novel antidiabetic drugs can include phytocompound(s) isolated from fruit plants that have demonstrated preclinical and clinical antidiabetic efficacy. However, over consumption of fruits with high levels of sugar content along with dried and canned fruits should be limited as despite the fact that plants are regarded as a safer and better alternative, excess amounts of those may welcome unwanted adverse effects. Further research is still recommended to identify and evaluate the precise molecular pathway of therapeutic actions revealed by these potential fruits.

## Figures and Tables

**Figure 1 molecules-27-08709-f001:**
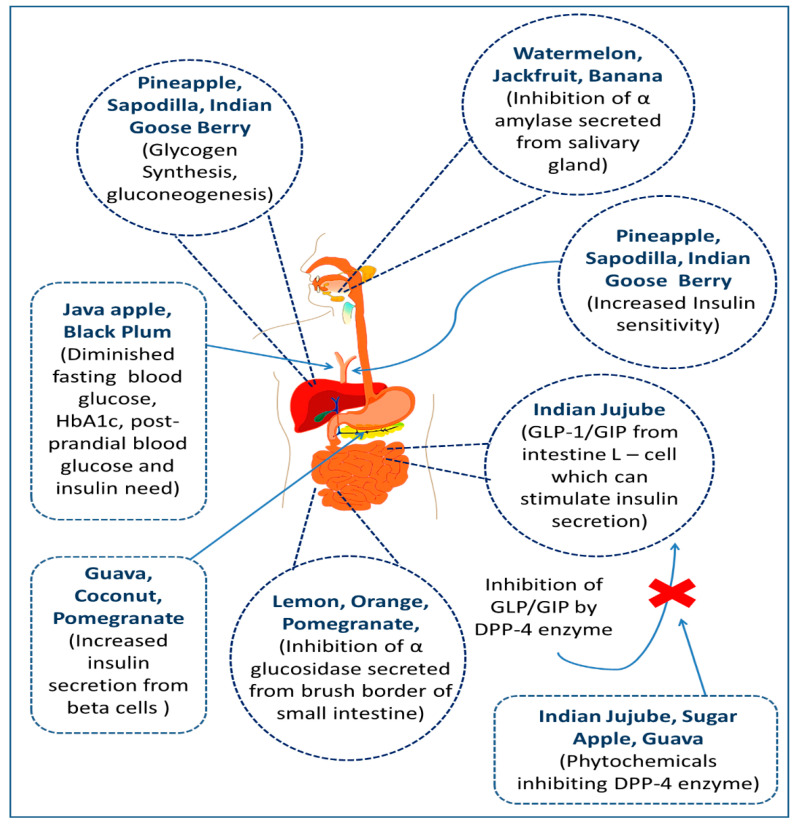
Mechanism of actions of fruit plants and reported phytochemicals to exert antidiabetic potentials.

**Figure 2 molecules-27-08709-f002:**
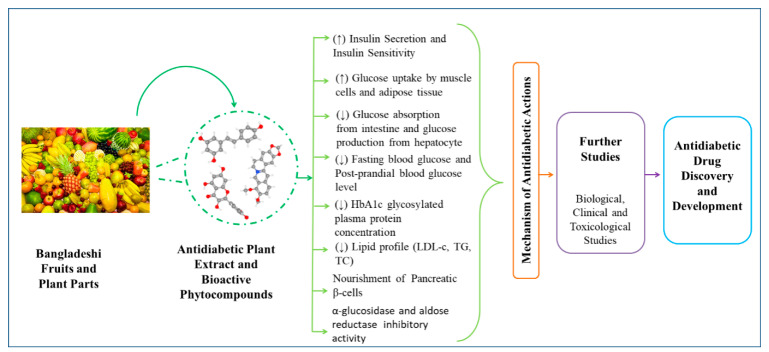
Commonly available Bangladeshi fruits and other plant parts for new antidiabetic drug discovery and development.

**Figure 3 molecules-27-08709-f003:**
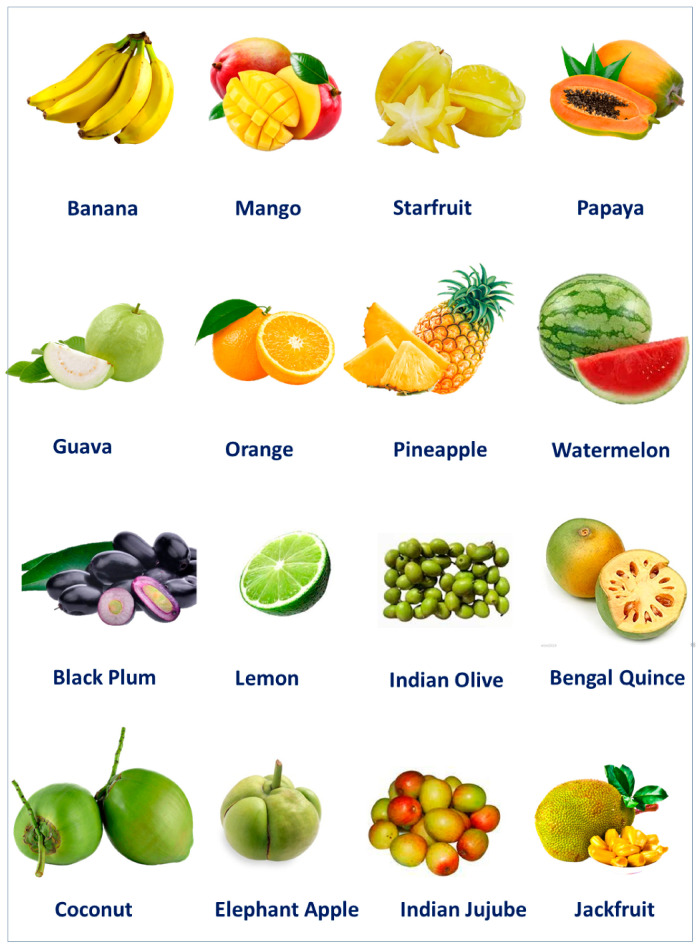
Few commonly available fruits in Bangladesh.

**Figure 4 molecules-27-08709-f004:**
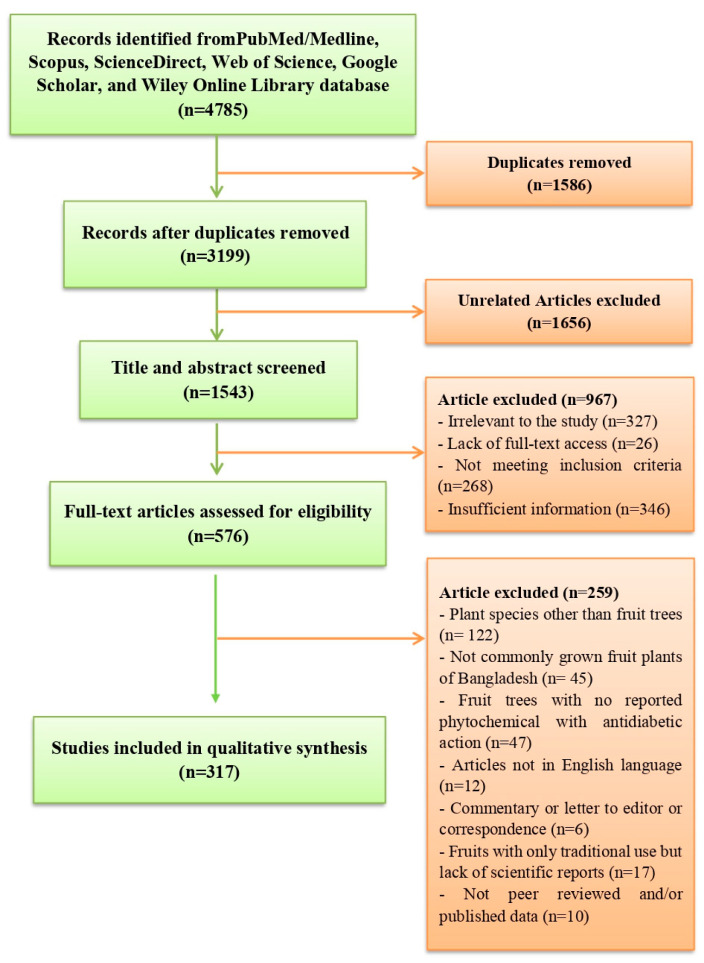
PRISMA for the identification and screening of articles to include in the study according to eligible criteria.

**Table 1 molecules-27-08709-t001:** Commonly available Bangladeshi fruit plants and their antidiabetic properties along with responsible phytochemicals and their modes of action.

S.L.	Species	Plant Part	Phytochemicals	Mode of Actions	References
**1**	*Aegle Marmelos*	Leaf	Aegeline	Blood glucose-lowering activity	[121]
			Aegelin 2; sitosterol; scopoletin	Blood glucose-lowering activity	[186]
**2**	*Annona squamosa*	Leaf	Rutin	Inhibition of α-glucosidase activity and improved insulin secretion	[187]
			Quercetin	Improving insulin secretion	[187,188]
			Isoquercetin	Inhibiting the activity of DPP-4	[187]
**3**	*Ananas comosus*	Fruit	Catechin	α-glucosidase inhibitory actions	[189,190]
			Epicatechin	Promoting β-cell regeneration	[190,191]
			Gallic acid	Improved glucose transporters and insulin sensitivity through PPAR-γ and Akt signaling	[190,192]
			*p*-coumaric acid	Lowering the blood glucose level, increasing the level of insulin	[190,193]
			Ferulic acid	Restoring blood glucose and serum insulin level; improving insulin sensitivity, hepatic glycogenesis, glucose tolerance, and insulin tolerance along with reducing the activity of glycogen synthase and glucokinase and also increasing activity of glycogen phosphorylase and enzymes of gluconeogenesis (PEPCK and G6Pase)	[190,194]
			Caffeic acid	Reducing blood glucose concentration by inhibiting the activity of glucose-6-phosphatase and increasing insulin secretion	[190,195]
			Ellagic acid	Stimulating insulin secretion and decreases glucose intolerance by acting on β-cells of the pancreas	[71,190]
			Vanillin	Reducing serum glucose level and increasing insulin level	[190,196]
**4**	*Artocarpus heterophyllus*	Whole Plant	Morin	Down-regulation of miR-29a expression level to improve insulin signaling and glucose metabolism	[197]
		Bark from the main trunk, Root	Betulinic acid	Activating AMPK, reducing blood glucose level, and stimulating mRNA expression of GLUT-4	[197,198]
		Root	Ursolic acid	Lowering blood glucose level	[197,199]
		Leaves, Stem, Root	β-sitosterol	Improving insulin resistance and insulin signaling along with reducing fasting blood glucose level and glycated hemoglobin	[197,200]
**5**	*Averrhoa carambola*	Fruit	2-dodecyl-6-methoxycyclohexa-2,5-diene-1, 4-dione	Improved blood sugar level and inhibition of the TLR-4/TGF-β signaling pathway	[156]
**6**	*Baccaurea motleyana*	Fruit	Limonene	Inhibiting protein glycation	[110,201]
			Carvacrol	α-amylase inhibitory actions	[110,201]
**7**	*Borassus* *flabellifer*	Fruit pulp	Tyrosol, Glucosyl-(6-1)-glycerol	α-glucosidase inhibitory actions	[202]
**8**	*Carica papaya*	Seed	Hexadecanoic acid, methyl ester; 11-octadecenoic acid, methyl ester; N, N-dimethyl-; n-hexadecanoic acid, and oleic acid	Inhibiting α-amylase and α-glucosidase	[132]
**9**	*Carissa carandas*	Leaf	Ursolic acid	Lowering blood glucose level	[203,204]
		Fruit	Myo-inositol	Improved insulin-stimulated glucose uptake in mature adipocytes; increased insulin sensitivity and glucose uptake	[36,205]
			β-amyrin	Improvement in blood sugar level and plasma insulin along with preservation of β cell integrity	[36,206]
**10**	*Citrullus lanatus*	Fruit	Citrulline	Improving insulin sensitivity	[166,207]
			6-gingerol	Lowering fasting blood glucose levels and improving glucose tolerance	[166,208]
			Arjunolic acid	Protective action on pancreatic β-cells, Reducing blood glucose level	[166,209]
			Lycopene	Reducing blood glucose level	[195]
**11**	*Citrus arunattifolia*	Fruit	Hesperetin	Reducing blood glucose levels and increasing plasma insulin concentration and glycogen levels, ameliorating the abnormality resulting from hyperglycemia in pancreatic β-cells, increasing the number of insulin immune-positive cells of the islets	[210,211]
			Quercetin	Increased glucose uptake via regulation of the AMPK pathway along with improved GLUT-4 expression and regeneration of β-cells in the pancreatic islets	[210,211]
			Rutin	Reducing blood glucose, elevating glycogen concentration	[210,211]
			Nobiletin	Enhancing glucose uptake via the PI3K/Akt signaling pathway	[210,211]
			Kaempferol	Inhibiting the activities of α-amylase and α-glucosidase along with glucose-lowering activity	[211,212]
			Naringenin	lowered the increased plasma glucose concentration, increased the activity of Superoxide dismutase (SOD)	[210,211]
			Apigenin	Reduction in serum glucose level, enhancing serum insulin level, nourishment of pancreatic β cell	[211,213,214]
**12**	*Citrus limon*	Fruit Peel	Nobiletin	Enhancing glucose uptake via the PI3K/Akt signaling pathway	[210,215]
		Fruit, leaf	Diosmin	Improvement in fasting plasma glucose concentrations, glycosylated hemoglobin (HbA1c), C-reactive protein (CRP), and the activities of Lipoprotein lipase (LPL) and lecithin cholesterol acyl transferase (LCAT) enzymes	[210]
		Fruit	Rhoifolin	Insulin mimetic activity	[210]
			Didymin	Improving insulin sensitivity, inhibitory action on α-glucosidase and diminished hepatic glucose production in insulin-resistant HepG2 cells	[210]
			Hesperidin	Reducing serum insulin and blood glucose level, normalizing the enzymatic activities of glucose-6-phosphatase and glucokinase enzyme	[210]
			Hesperetin	Reducing blood glucose levels and increasing plasma insulin concentration and glycogen levels, ameliorating the abnormality resulting from hyperglycemia in pancreatic β-cells, increasing the number of insulin immune-positive cells of the islets	[210]
			Rutin	Reducing blood glucose, elevating glycogen concentration	[210]
			Quercetin	Increased glucose uptake via regulation of the AMPK pathway along with improved GLUT-4 expression and regeneration of β-cells in the pancreatic islets	[210]
**13**	*Citrus maxima*	Fruit	β-sitosterol	Improving insulin resistance and insulin signaling along with reducing fasting blood glucose level and glycated hemoglobin	[200,216]
			Naringenin	Increased glucose uptake	[216,217]
			Naringin	Improved serum insulin level; enhanced mRNA expression of insulin receptor β-subunit and GLUT-4	[216,218]
**14**	*Citrus reticulata*	Fruit	alpha-pinene	Decreasing fasting blood glucoselevels	[219,220]
			Limonene	Inhibiting protein glycation	[201,219]
			Rutin	Reducing blood glucose, elevating glycogen concentration	[210]
			Quercetin	Increased glucose uptake via regulation of the AMPK pathway along with improved GLUT-4 expression and regeneration of β-cells in the pancreatic islets	[210]
			Naringin	Modifying the release of insulin from isolated islet and intestinal glucose absorption, improved expression of GLUT-4 in adipose tissue, and free circulating glucose uptake from the blood to peripheral tissues	[210,221]
			Naringenin	Lowered the increased plasma glucose concentration, increased the activity of Superoxide dismutase (SOD)	[210]
			Didymin	Improving insulin sensitivity, inhibitory action on α-glucosidase and diminished hepatic glucose production in insulin-resistant HepG2 cells	[210]
			Hesperidin	Reducing serum insulin and blood glucose level, normalizing the enzymatic activities of glucose-6-phosphatase and glucokinase enzyme	[210]
			8-Prenylnaringenin	Regulating the expression of Galectin-3 (Gal-3) protein which is overexpressed during the diabetic state, promoting glycation end products (AGEs) production	[210]
		Fruits and leaves	Diosmin	Improvement in fasting plasma glucose concentrations, glycosylated hemoglobin (HbA1c), C-reactive protein (CRP), and the activities of Lipoprotein lipase (LPL) and lecithin cholesterol acyl transferase (LCAT) enzymes	[210]
		Fruit peel	Nobiletin	Enhancing glucose uptake via the PI3K/Akt signaling pathway	[210]
			Tangeretin	Increasing glucose uptake, improving glycogen level and the activities of glycogen synthase and glycogen phosphorylase, regeneration of pancreatic β-cells in the islets	[210]
		Dried pericarp	Hesperetin	Reducing blood glucose levels and increasing plasma insulin concentration and glycogen levels, ameliorating the abnormality resulting from hyperglycemia in pancreatic β-cells, increasing the number of insulin immune-positive cells of the islets	[210,222]
		Albedo of fruit	Neohesperidin	Improved glucose tolerance and insulin sensitivity along with abatement in the blood glucose level	[210,223]
**15**	*Cocos nucifera*	Coconut water	Myo-inositol	Improved insulin-stimulated glucose uptake in mature adipocytes; increased insulin sensitivity and glucose uptake	[205,224]
**16**	*Cucumis melo*	Seed	Gallic acid	Improved glucose transporters and insulin sensitivity through PPAR-γ and Akt signaling	[192,225]
			Vanillic acid	Translocation of GLUT-4 via the AMPK-dependent pathway	[225,226]
			4-Hydroxybenzoic acid	Increasing serum insulin levels and liver glycogen content	[225,227]
**17**	*Dillenia indica*	Leaf	3,5,7,-Trihydroxy-2-(4-hydroxy-benzyl)-chroman-4-one	Enhanced serum insulin level and reduced fasting blood glucose level	[228]
			Betulinic acid, Quercetin, β-sitosterol, Stigmasterol	Inhibiting activities of α-amylase and α-glucosidase	[191]
**18**	*Diospyros malabarica*	Leaf	Betulinic acid	Activating AMPK, reducing blood glucose level, and stimulating mRNA expression of GLUT-4	[86,198]
		Fruit, Leaf	β-sitosterol	Improving insulin resistance and insulin signaling along with reducing fasting blood glucose level and glycated hemoglobin	[86,200]
**19**	*Elaeocarpus floribundus*	Leaf	Myricitrin	α-amylase inhibitory actions	[229]
**20**	*Limonia acidissima*	Whole Plant	Stigmasterol	lowering serum glucose level	[230,231]
			Umbelliferone	Improving glucose uptake	[231,232]
			Lupeol	Improving serum insulin level, reducing glycated hemoglobin, serum glucose	[230,231]
			β-sitosterol	Improving insulin resistance and insulin signaling along with reducing fasting blood glucose level and glycated hemoglobin	[200,231]
**21**	*Litchi chinensis*	Fruit	Epicatechin	Promoting β-cell regeneration, enhancement of glucose consumption in HepG2 cells	[122]
		Seed	(2R)-naringenin-7-O-(3-O-α-L-rhamnopyranosyl-β -D-glucopyranoside); narirutin; quercetin; phlorizin	dose-dependent α-glucosidase inhibitory actions	
		Fruit	Oligonol	Ameliorating insulin resistance	
		Flower	Gentisic acid	Inhibiting α-amylase and α-glucosidase	[233,234]
			Epicatechin	Promoting β-cell regeneration, enhancement of glucose consumption in HepG2 cells	[122,233,235]
**22**	*Mangifera indica*	Kernel	6-O-galloyl-5′-hydroxy mangiferin; mangiferin; 5-hydroxy mangiferin; methyl gallate	Reducing the blood glucose levels	[236]
		Leaf	1,2,3,4,6 Penta-O-galloyl-β-d-glucose	Inhibiting 11-β-HSD-1 and ameliorating high-fat diet-induced diabetes	[237]
**23**	*Manilkara zapota*	Fruit peel	Gallic acid	Improved glucose transporters and insulin sensitivity through PPAR-γ and Akt signaling	[192,238]
			Ellagic acid	Stimulating insulin secretion and decreases glucose intolerance by acting on β-cells of the pancreas	[71,238]
			Ferulic acid	Restoring blood glucose and serum insulin level; improving insulin sensitivity, hepatic glycogenesis, glucose tolerance, and insulin tolerance along with reducing the activity of glycogen synthase and glucokinase and also increasing activity of glycogen phosphorylase and enzymes of gluconeogenesis (PEPCK and G6Pase)	[194,238]
			Catechin	α-glucosidase inhibitory actions	[189,238]
			Epicatechin	Promoting β-cell regeneration	[235,238]
			Quercetin	Improving insulin secretion	[188,238]
**24**	*Musa sapientum*	Pulp	Quercetin	Improving insulin secretion	[188,239]
			Pectin	Reducing blood glucose level, improving liver glycogen	[188,239]
		Leaf	β-amyrin	Improvement in blood sugar level and plasma insulin along with preservation of β-cell integrity	[206,239]
			Lanosterol	Reducing blood glucose level	[239,240]
**25**	*Phyllanthus embelica*	Fruit	Coliragin	Regulating fasting blood glucose, plasma insulin, and glycated hemoglobin (HbA1c) level	[241,242]
			Gallic acid	Improved glucose transporters and insulin sensitivity through PPAR-γ and Akt signaling	[192,242]
			Ellagic acid	Stimulating insulin secretion and decreases glucose intolerance by acting on β-cells of the pancreas	[71,242]
			Cinnamic Acid	Stimulation of insulin secretion, enhancement of insulin signaling pathway, the activity of pancreatic β-cell and glucose uptake, inhibition of hepatic gluconeogenesis, protein glycation, and insulin fibrillation along with the delay of carbohydrate digestion and glucose absorption	[243,244]
**26**	*Psidium* *guajava*	Leaf	Guaijaverin	Inhibiting the activity of DPP-4	[245]
			Avicularin	Inhibiting glucose uptake mediated by GLUT-4	
**27**	*Punica granatum*	Rind	Valoneic acid dilactone	Inhibiting the activity of aldose reductase and α-amylase enzyme, ameliorating blood glucose levels by inhibiting protein tyrosine phosphatase 1B (PTP1B), improved insulin secretion from pancreatic β cells or its release from the bound form along with insulin-mimetic actions along with rejuvenation of glucose utilization technique	[149]
		Fruit	Caffeic acid	Reducing blood glucose concentration by inhibiting the activity of glucose-6-phosphatase and increasing insulin secretion	[246,247]
		Fruit, pericarp	Quercetin	Increased glucose uptake via regulation of the AMPK pathway along with improved GLUT-4 expression and regeneration of β-cells in the pancreatic islets	[210,247]
			Rutin	Reducing blood glucose, elevating glycogen concentration	[210,212,247]
			Catechin	α-glucosidase inhibitory actions	[189,247]
		Fruit, rind, flower	Gallic acid	Improved glucose transporters and insulin sensitivity through PPAR-γ and Akt signaling	[192,247]
		Fruit, seed, flower	Ellagic acid	Stimulating insulin secretion and decreases glucose intolerance by acting on β-cells of the pancreas	[71,247,248]
		Seed	Punicic acid	Increasing serum insulin levels and modulating glucose homeostasis	[247,249]
		Flower	Tricetin 4′-O-β-glucopyranoside	Inhibiting activities of α-amylase and α-glucosidase	[248]
			Tricetin	Inhibiting activities of α-amylase and α-glucosidase	[248]
			Ursolic acid	Lowering blood glucose level	[197,247]
			Mascilinic acid	Improving insulin resistance, hepatic glycogen content, and inhibiting glycogen phosphorylase activity in HepG2 cells	[247,250]
			Luteolin	Regulating blood glucose, HbA1c, and insulin level	[248,251]
		Fruit, Leaf, Root, Bark	Punicalin	α-glucosidase inhibitory actions	[247,252]
		Leaf	Apigenin	Reduction in serum glucose level, enhancing serum insulin level, nourishment of pancreatic β-cell	[213,222,247]
		Fruit, Root, bark	Punicalagin	α-glucosidase inhibitory actions	[247,252]
**28**	*Spondias mombin*	Leaf	3β-olean-12-en-3-yl (9Z)-hexadec-9-enoate	α-amylase inhibitory actions	[253]
			Mombintane I; mombintane II	Glucose lowering and α-amylase inhibitory actions	[254]
**29**	*Syzygium cumini*	Leaf	Stigmasterol	Lowering serum glucose level	[230]
			Lupeol	Improving serum insulin level, reducing glycated hemoglobin and serum glucose	
			β-sitosterol	Improving insulin resistance and insulin signaling along with reducing fasting blood glucose level and glycated hemoglobin	[200,230]
**30**	*Syzygium samarangense*	Leaf	2′,4′-dihydroxy-3′, 5′-dimethyl-6′-methoxychalcone	Lowering blood glucose level	[255]
		Fruit	Vescalagin	Improving insulin resistance and glycemic metabolism	[256]
**31**	*Tamarindus indica*	Leaf	Limonene	Inhibiting protein glycation	[201,257]
		Root bark	Pinitol	Insulin mimetic activity and regulating glucose uptake	[257,258]
		Seed	β-amyrin	Improvement in blood sugar level and plasma insulin along with preservation of β-cell integrity	[206,257]
		Seed, Root bark	β-sitosterol	Improving insulin resistance and insulin signaling along with reducing fasting blood glucose level and glycated hemoglobin	[200,257]
		Fruit	Lupeol	Improving serum insulin level, reducing glycated hemoglobin, serum glucose	[230,257]
			Naringenin	Increased glucose uptake	[217,257]
			Catechin	α-glucosidase inhibitory actions	[189,257]
			Epicatechin	Promoting β-cell regeneration	[235,257]
**32**	*Ziziphus mauritiana*	Aerial Part	Berberine	Insulin mimetic activity and improving the action of insulin by activating AMPK, decreasing insulin resistance through protein kinase C-dependent up-regulation of insulin receptor expression, enhancing GLP-1 secretion and modulating its release, inhibiting actions of DPP-4	[203]
			Quercetin	Improving insulin secretion	[188,203]
			Diosgenin	Reducing intestinal disaccharidases, blood glucose level, and activity of α-Glucosidase along with modification of hepatocyte absorption of glucose and insulin resistance and insulin secretion	[203,259]
			Kaempferol	Inhibiting the activities of α-amylase and α-glucosidase along with glucose-lowering activity	[203,212]
		Fruit	Ursolic acid	Lowering blood glucose level	[197,203]
			Oleanolic acid	Improved blood glucose and serum insulin levels	[203,260]
		Leaf	Swertisin	Generating new β-cells	[203,261]

**Table 2 molecules-27-08709-t002:** Chemical class-wise prospective antidiabetic phytochemicals found in fruit plants.

Alkaloids	Coumarins	Flavonoids	Polyphenols and Phenolic Compounds	Terpenes and Triterpenoids	Fatty Acids	Tannins and Ellagitannins	Xanthones	Miscellaneous
Aegelin 2	Scopoletin	Swertisin	Ferulic acid	Ursolic acid	n-hexadecanoic acid	Punicalagin	5-hydroxy mangiferin	Pinitol (Cyclic polyol)
Aegeline	Umbelliferon	Apigenin	Gallic acid	Limonene	Cinnamic Acid	Punicalin	6-O-galloyl-5′-hydroxy mangiferin	Neohesperidin (Glycoside)
Berberine		Avicularin	Gentisic acid	Arjunolic acid	Oleic acid	Vescalagin	Mangiferin	Citrulline (Non-essential amino acid)
		Catechin	p-coumaric acid	Betulinic acid	Punicic acid	Punicalagin		Diosgenin (Steroidal sapogenin)
		Didymin	Carvacrol	Lanosterol		Valoneic acid dilactone		
		Diosmin	Methyl gallate	Lupeol				
		Tangeretin	Vanillic acid	Mascilinic acid				
		Epicatechin	Vanillin	Oleanolic acid				
		Guaijaverin	Mombintane I	β-amyrin				
		Hesperetin	Mombintane II	α-Pinene				
		Hesperidin	Sitosterol					
		Isoquercetin	Stigmasterol					
		Kaempferol	β-sitosterol					
		Luteolin	Phlorizin					
		Morin	6-gingerol					
		Myricitrin	Caffeic acid					
		Naringenin	Oligonol					
		Naringin	Ellagic Acid					
		Nobiletin						
		Quercetin						
		Rhoifolin						
		Rutin						
		8-Prenylnaringenin						

## Data Availability

Not applicable.

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
