# Peer review of "Antidiabetic Potential of Commonly Available Fruit Plants in Bangladesh: Updates on Prospective Phytochemicals and Their Reported MoAs"

_molecules, 2022, doi:10.3390/molecules27248709_

Round 1

Reviewer 1 Report

-The abstract can be revised into a more concise manner.

-There are a lot of review papers about fruit plants phytochemicals. Some of the antidiabetic potential of fruit plants in Bangladesh has been reported. The authors must think about the needs of the reader and fill into the gap of the field of knowledge.

-Ref 10, 12, 266 self citations detected. More relevant citations can be used to replaced the self citations to provide a more comprehensive reading experience to the readers. 

-Table 1 and Table 3 information can be compiled in 1 table. 

-Figure 4 - What is the purpose of listing down all the chemical structure of phytochemicals associated with antidiabetic potentials?

Author Response

Authors’ Reply to Reviewers’ Comments

We are thankful to the to the reviewers for their scholarly comments. We have addressed all the concerned queries to the point raised by the reviewer and revised our manuscript accordingly. However, we have also edited he English language of the manuscript by a native speaker as well as online grammar tool Grammarly. Kindly find our responses towards the reviewer’s comment mentioned below:

Reviewer’s Comment 1: The abstract can be revised into a more concise manner.

Authors’ Response 1: We would like to thank to the Reviewer for his/her scholarly observation. We have rejuvenated the Abstract as per his suggestion.

Reviewer’s Comment 2: There are a lot of review papers about fruit plants phytochemicals. Some of the antidiabetic potential of fruit plants in Bangladesh has been reported. The authors must think about the needs of the reader and fill into the gap of the field of knowledge.

Authors’ Response 2: We have mentioned the utility and b=novelty of our present work in the discussion section to focus on gap fill up of the existing field (Lines 983-990)

Reviewer’s Comment 3: Ref 10, 12, 266 self citations detected. More relevant citations can be used to replaced the self citations to provide a more comprehensive reading experience to the readers. 

Authors’ Response 3: We have replaced Ref 10 ad 12. But ref 26 is really important for our review work as it focuses on the recent trend in antidiabetic drugs’ status. This paper is published in 2022 in Frontiers in Endocrinology.

Reviewer’s Comment 4: Table 1 and Table 3 information can be compiled in 1 table. 

Authors’ Response 4: These two tables are merged as per the suggestion.

Reviewer’s Comment 5: Figure 4 - What is the purpose of listing down all the chemical structure of phytochemicals associated with antidiabetic potentials?

Authors’ Response 5: We have removed figure 4 from the revised manuscript.

Reviewer 2 Report

I have review the manuscript entitled Antidiabetic potential of commonly available fruit plants in Bangladesh: updates on prospective phytochemicals along with their reported mechanism of actions

Abstract

The sentence from line 36 to 40 should be removed from the abstract. Line 49 seeds, leaves, and bark (the period should be removed).

Introduction

Line 66: 1,29000 what does this number mean?

Line 92,93: The word "blessing" should be replaced with a more appropriate word.

Line 94 to 96: I suggest quoting this article: https://doi.org/10.3390/horticulturae7080212

Line 115 to line 118: These two sentences are repeated in almost every paragraph, please reword or delete them

line 132: Also dried fruit (dried should be lower case).

The entire paragraph from line 115 to 139 should be rewritten and made into flowing text because it is now too simple and almost every sentence is the same

Results

4.1. banana (Latin names should be written in italics...)

Line 152: The Latin name of the banana should be in italics... please check all Latin names to be written in italics.

Line 160: The Latin name of Bengal quince should be in italics...

line 175: the dates should all be written with the same decimal places (not 15 +- 0.54), but 15.00 +- 0.54...

Line 181: again Latin names in italics... Also, every paragraph says the same thing: it's about phytochemicals like phenolic acids, steroids, alkaloids, polysaccharides, flavonoids, essential oils, and lignans. I would suggest making a paragraph stating that all fruits studied are a rich source of primary and secondary metabolites... instead of repeating the same thing over and over

Table 1 should be more condensed and clear, because it is hard to see which fruit contains which chemical and what effects it has on human health...

Line 840 alpha-glucoside is correct alpha-glucoside?? the same is true for alpha-amilaze

Line 987: the word blessed should be chagned.

The authors should rewrite the manuscript taking into account my comments. The current presentation of the manuscript is not suitable for publication.

Author Response

Authors’ Reply to Reviewers’ Comments

We are thankful to the to the reviewers for their scholarly comments. We have addressed all the concerned queries to the point raised by the reviewer and revised our manuscript accordingly. However, we have also edited he English language of the manuscript by a native speaker as well as online grammar tool Grammarly. Kindly find our responses towards the reviewer’s comment mentioned below:

Abstract

Reviewer’s Comment 1: The sentence from line 36 to 40 should be removed from the abstract. Line 49 seeds, leaves, and bark (the period should be removed).

Authors’ Response 1: We would like to thank to the Reviewer for his/her scholarly observation. We have rejuvenated the Abstract as per his suggestion.

Introduction

Reviewer’s Comment 2: Line 66: 1,29000 what does this number mean?

Authors’ Response 2: The number would be 1,29,000. We have revised it in the manuscript.

Reviewer’s Comment 3: Line 92,93: The word "blessing" should be replaced with a more appropriate word.

Authors’ Response 3: The manuscript is revised as per the scholarly comment of reviewer and marked in yellow color.

Reviewer’s Comment 4: Line 94 to 96: I suggest quoting this article: https://doi.org/10.3390/horticulturae7080212

Authors’ Response 4: We have cited this contemporary article in our paper to upgrade the overall quality of existing manuscript.

Reviewer’s Comment 5: Line 115 to line 118: These two sentences are repeated in almost every paragraph, please reword or delete them

Authors’ Response 5: The manuscript is revised as per the scholarly comment of reviewer and marked in yellow color.

Reviewer’s Comment 6: line 132: Also dried fruit (dried should be lower case).

Authors’ Response 6: We have adjusted the error in the revised manuscript as per the suggestion.

Reviewer’s Comment 7: The entire paragraph from line 115 to 139 should be rewritten and made into flowing text because it is now too simple and almost every sentence is the same

Authors’ Response 7: We have revised the error in the revised manuscript as per the suggestion to make it concise and easy to understand for the readers.

Results

Reviewer’s Comment 8: 4.1. banana (Latin names should be written in italics...)

Authors’ Response 8: We are thanking reviewer for the scholarly and very crucial observation. Every scientific name throughout the manuscript have been revised in the revised manuscript to maintain the integrity of scientific writing.

Reviewer’s Comment 9: Line 152: The Latin name of the banana should be in italics... please check all Latin names to be written in italics.

Authors’ Response 9: Every scientific name throughout the manuscript have been revised in the revised manuscript to maintain the integrity of scientific writing.

Reviewer’s Comment 10: Line 160: The Latin name of Bengal quince should be in italics...

Authors’ Response 10: Every scientific name throughout the manuscript have been revised in the revised manuscript to maintain the integrity of scientific writing.

Reviewer’s Comment 11: line 175: the dates should all be written with the same decimal places (not 15 +- 0.54), but 15.00 +- 0.54...

Authors’ Response 11: The issue is revised as per the scholarly suggestion of the reviewer.

Reviewer’s Comment 12: Line 181: again Latin names in italics... Also, every paragraph says the same thing: it's about phytochemicals like phenolic acids, steroids, alkaloids, polysaccharides, flavonoids, essential oils, and lignans. I would suggest making a paragraph stating that all fruits studied are a rich source of primary and secondary metabolites... instead of repeating the same thing over and over.

Authors’ Response 12: We have revised the issue in lie 181. But we kept few chemical classes in some fruits as all fruits do not contain all kind og phytochemical classes. So we are requesting the reviewer to consider the issue.

Reviewer’s Comment 13: Table 1 should be more condensed and clear, because it is hard to see which fruit contains which chemical and what effects it has on human health...

Authors’ Response 13: Table 1 has been revised as per the suggestion in the revised manuscript.

Reviewer’s Comment 14: Line 840 alpha-glucoside is correct alpha-glucoside?? the same is true for alpha-amilaze

Authors’ Response 14: We are thanking reviewer for the comment. A dedicated point has been written entitled “Inhibition of α-amylase secreted from salivary gland” (Lines 837-841).

Reviewer’s Comment 15: Line 987: the word blessed should be changed.

Authors’ Response 15: The word has been rewritten as per the suggestion.

Round 2

Reviewer 2 Report

Line 61: The number should be written like 129.000 or 129000.

Chapter 4.21 Orange

The numbers should be written on two decimal places (recheck the entire mansucript)

The μL/kg has liters written as L and sometiems the liters are written as small l. Please make the entire manuscript the same with liters being marked as L. This also goes for grams (should be small g), kilograms etc.

Otherwise the manuscript is now improved.